# AIOS: LLM Agent Operating System

**Kai Mei**          **Xi Zhu**          **Wujiang Xu**          **Mingyu Jin**          **Wenyue Hua**

**Zelong Li**     **Shuyuan Xu**     **Ruosong Ye**     **Yingqiang Ge**     **Yongfeng Zhang**

Department of Computer Science, Rutgers University[*]

## Abstract

LLM-based intelligent agents face significant deployment challenges, particularly related to resource management. Allowing unrestricted access to LLM or tool resources can lead to inefficient or even potentially harmful resource allocation and utilization for agents. Furthermore, the absence of proper scheduling and resource management mechanisms in current agent designs hinders concurrent processing and limits overall system efficiency. To address these challenges, this paper proposes the architecture of AIOS (LLM-based AI Agent Operating System) under the context of managing LLM-based agents. It introduces a novel architecture for serving LLM-based agents by isolating resources and LLM-specific services from agent applications into an AIOS kernel. This AIOS kernel provides fundamental services (e.g., scheduling, context management, memory management, storage management, access control) for runtime agents. To enhance usability, AIOS also includes an AIOS SDK, a comprehensive suite of APIs designed for utilizing functionalities provided by the AIOS kernel. Experimental results demonstrate that using AIOS can achieve up to $2.1\times$ faster execution for serving agents built by various agent frameworks. The source code is available at https://github.com/agiresearch/AIOS.

## 1 Introduction

In the field of autonomous agents, research efforts (Wooldridge & Jennings, 1995; Jennings et al., 1998; Bresciani et al., 2004) are made towards agents that can perceive environments, understand instructions, make decisions, take action and learn from feedbacks. The advent of large language models (LLMs) (Achiam et al., 2023; Touvron et al., 2023a; Team et al., 2023) has brought new possibilities to the agent development Ge et al. (2023a). Current LLMs have shown great power in understanding instructions (Ouyang et al., 2022; Chung et al., 2022; Touvron et al., 2023b; Geng et al., 2022), reasoning and solving problems (Kojima et al., 2022; Nijkamp et al., 2022; Taylor et al., 2022; Hao et al., 2023; Kim et al., 2023), and interacting with human users (Ross et al., 2023) as well as external environments (Driess et al., 2023; Brohan et al., 2023). Built upon these powerful LLMs, emergent LLM-based agents (Ge et al., 2023a; Yao et al., 2023; Shinn et al., 2023; Deng et al., 2023; Packer et al., 2023; Wu et al., 2024) can present strong task fulfillment abilities in diverse environments, ranging from virtual assistants to more sophisticated reasoning and problem-solving systems.

An illustrative example of an LLM-based agent's real-world task execution is demonstrated in Figure 1, where a travel agent processes a trip organization request. The agent methodically decomposes this request into executable steps—booking flights, reserving accommodations, processing payments, and updating calendars according to user preferences. Throughout execution, the agent exhibits reasoning and decision-making capabilities derived from its LLM foundation, distinguishing it from traditional applications constrained by predetermined functions or workflows. Implementing this travel scenario requires the agent to seamlessly integrate LLM-related services (preference retrieval, API selection, response generation) with conventional OS services (disk access, software execution).

---

[*]Author emails: {kai.mei, xi.zhu, wujiang.xu, mingyu.jin, wenyue.hua, zelong.li, shuyuan.xu, ruosong.ye, yingqiang.ge, yongfeng.zhang}@rutgers.edu

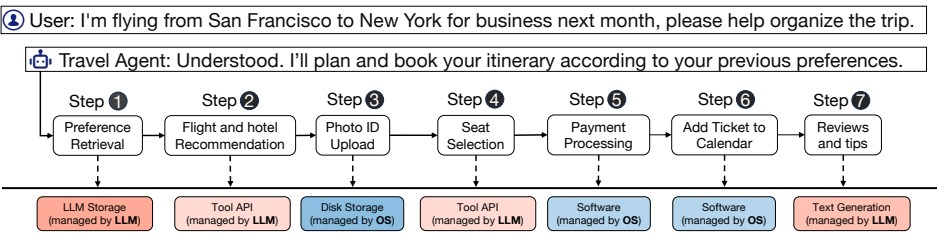

Figure 1: A motivating example of how an agent (i.e., travel agent) requires both LLM-related and Non-LLM-related (i.e., OS) services to complete a task, where color in red represents services related to LLM and color in blue represents services not related to LLM.

Current agent frameworks exhibit critical design limitations by granting agents direct access to system-level resources like LLMs and tools (Qin et al., 2024), compromising resource optimization and introducing potential vulnerabilities. Without proper scheduling, agents can monopolize resources such as flooding the LLM with requests while others wait. The absence of effective resource management mechanisms significantly impairs system efficiency. Under concurrent conditions, existing frameworks (e.g., Autogen, Langchain) employ an inefficient trial-and-error approach for LLM calls: prompts are converted to tensors and loaded into GPU memory until CUDA memory limits trigger exceptions, forcing tensor deallocation and requiring multiple retry attempts, which substantially degrades system throughput in scenarios where numerous agents compete for limited LLM resources.

To mitigate above-mentioned limitations, we introduce AIOS, an architecture designed to serve LLM-based agents more efficiently. Our contributions are as below.

○ *New Agent-serving Architecture.* We introduce AIOS, a novel LLM-based agent serving architecture. AIOS divides agent applications and resources for agents such as LLMs and tools into distinct layers, i.e., the application layer and the kernel layer. This separation facilitates systematic resource management, efficiency optimization, and safety enhancement.

○ *AIOS Kernel Design and Implementation.* At the core of AIOS, we design and implement an AIOS kernel to encapsulate resource management abstractions. In this kernel, agent queries are decomposed into sub execution units (i.e., AIOS syscalls) to facilitate parallelism. We design an agent scheduler to orchestrate syscall execution across modules, while memory, storage, and tool managers and the LLM core are responsible for handling dispatched syscalls. We also design the context manager to handle context interruption and design the access manager to verify agent operations to ensure the reliability of the AIOS kernel.

○ *AIOS SDK Development.* We develop the AIOS SDK, which provides a higher level abstraction of kernel functionalities, allowing developers to focus on application logic and higher-level functionalities without being burdened by complicated kernel details.

○ *Empirical Results.* We conduct extensive evaluations of AIOS on agents developed using various agent frameworks. The results demonstrate that AIOS can maintain the performance of agents across standard agent benchmarks and can even enhance performance in benchmarks that involve tool calling. Furthermore, AIOS significantly improves execution efficiency, achieving up to a 2.1× increase in execution speed for serving agents across different frameworks. These experimental results underscore the effectiveness of AIOS in optimizing both agent performance and execution speed in serving agents.

## 2 The Architecture of AIOS

As depicted in Figure 2, the AIOS architecture is divided into three distinct layers: the application, kernel, and hardware layers. This layered design is intended to establish a clear separation of concerns within the system. Higher-level applications abstract the complexities of the underlying layers, interacting with them through well-defined interfaces such as software development kits (SDKs) and system calls.

**Application Layer.** The application layer leverages the AIOS SDK, providing interfaces for requesting system resources within AIOS. This design relieves agents from resource

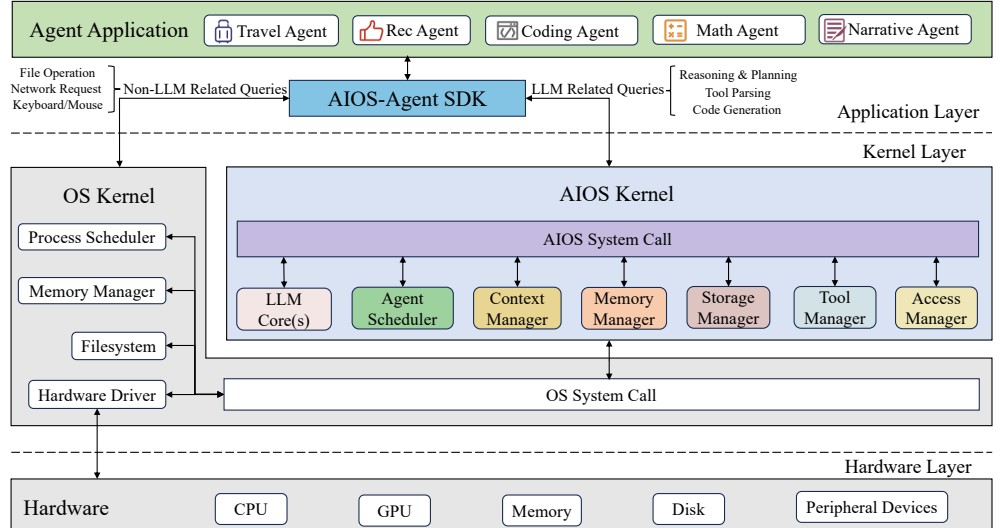

Figure 2: An overview of the AIOS architecture of distinct layers. Application layer facilitates the design and development of agent applications. Kernel layer manages core functionalities and resources to serve agent applications. Hardware layer controls and manages physical computing resources and devices to support kernel layer functionalities.

management while enforcing isolation by preventing direct resource manipulation. The AIOS SDK supports both native agent development and non-native agents adapted from diverse frameworks including ReAct (Yao et al., 2023), Reflexion (Shinn et al., 2023), Autogen (Wu et al., 2023), Open-Interpreter (Lucas, 2024), and MetaGPT (Hong et al., 2023). Non-native agents interact with AIOS kernel resources through adapter functions, while native development is streamlined via pre-defined APIs that invoke system calls. This abstraction allows developers to focus on agent logic rather than resource management details.

**Kernel Layer.**    The kernel layer integrates two components: the traditional OS kernel for non-LLM computing tasks and our innovative AIOS kernel. Within the AIOS kernel, specialized modules process agent requests through system calls. A scheduler dispatches these calls to appropriate modules using advanced scheduling strategies (detailed in Section 3.3). We design a unified interface encapsulating LLMs as cores, akin to CPU cores, enabling integration of diverse LLM endpoints. To support LLM context switching, we implement a context manager with snapshot and restoration capabilities (Section 3.4). For efficient agent data handling, we develop a memory manager for runtime operations (Section 3.5) and a storage manager for persistent storage (Section 3.6). Additionally, a tool manager loads tools and resolves call conflicts for AIOS SDK supported tools (Section 3.7), while an access manager implements access control and user intervention protocols (Section 3.8).

**Hardware Layer.** The hardware layer consists of the physical components of the system, such as the CPU, GPU, memory, disk, and peripheral devices. The hardware layer is not the main focus of the work—AIOS kernel does not directly interact with the hardware but relies on OS system calls to access the physical resources in the hardware layer.

# 3   AIOS Kernel

In this section, we start with an overview of the AIOS kernel, highlighting how each module collaborates with other modules to support integrated functionalities. Following this, we provide an in-depth look into the design and implementation of each module, discussing their roles and contributions to the overall AIOS architecture.

## 3.1   Relationship and Connection between Modules

Within the AIOS kernel, agent queries are decomposed into categorized system calls (LLM processing, memory access, storage operations, tool usage) as shown in Figure 3, with a comprehensive syscall catalog available in Appendix A.1. Each syscall is thread-bound

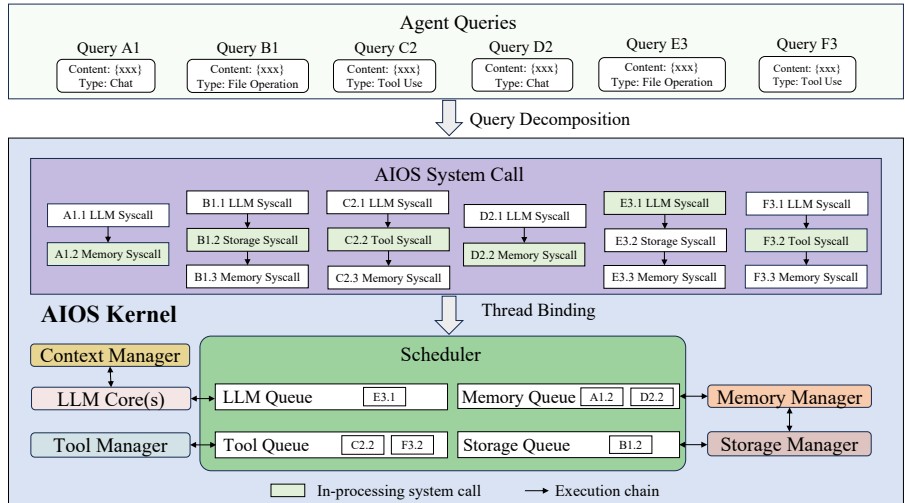

Figure 3: How agent queries are decomposed into AIOS system calls and how AIOS system calls are dispatched and scheduled. We omit the access manager module here as the access-related system calls will not be dispatched by the scheduler.

and dispatched by the scheduler, which centralizes queue management across all modules. Syscalls are routed to appropriate module queues based on their attribute sets, with each module monitoring its designated queue for scheduled calls. Context manager will be triggered within LLM core for handling context interruptions instead of being scheduled.

## 3.2 LLM Core

Due to the various deployment options of LLMs, e.g., which LLM is used, whether the LLM is hosted on cloud or on local device, what hardware conditions the LLM requires, or which inference framework is used, we encapsulate each LLM instance adopting different deployment options as a core, akin to a CPU core in a traditional operating system. This design allows us to treat each LLM instance as a dedicated processing unit, enhancing the modularity and extensibility within the AIOS architecture. To accommodate different LLM instances, we introduce a wrapper for each LLM instance and design unified system calls within this wrapper specifically for LLM inference. By abstracting an LLM instance as a core and implementing standardized system calls, AIOS provides a flexible way to integrate LLM instances under different deployment options, attributed to the modular design of the LLM core. Detailed information of LLM core is provided in Appendix A.2.

## 3.3 Scheduler

We centralize all queues within the scheduler module rather than distributing them across processing modules, isolating request management responsibilities and allowing each module to focus solely on execution. This centralization simplifies cross-module task coordination and provides a unified scheduling framework. For managing system calls, we implement two classic algorithms: First-In-First-Out (FIFO), which processes calls in arrival order but may increase waiting times for later requests, and Round Robin (RR), which cycles through calls in time-sliced fashion for more balanced resource distribution. The RR strategy is supported by our context interrupt mechanism for LLM inference, detailed in Section A.3.

## 3.4 Context Manager

LLM inference time creates bottlenecks through long-running system calls that monopolize resources. Our context interrupt mechanism addresses this via task interruption and resumption through snapshot and restoration operations. The context manager designs two approaches: text-based (for closed-source LLMs without logits access, saving decoded outputs) and logits-based (preserving intermediate search tree structure for finer-grained state restoration). The logits-based method is illustrated in Figure 4. Using beam search (common in LLMs (Touvron et al., 2023b; Jiang et al., 2023; Biderman et al., 2023)), with beam width 1 for simplicity, we demonstrate the process: When processing *Determine whether there*

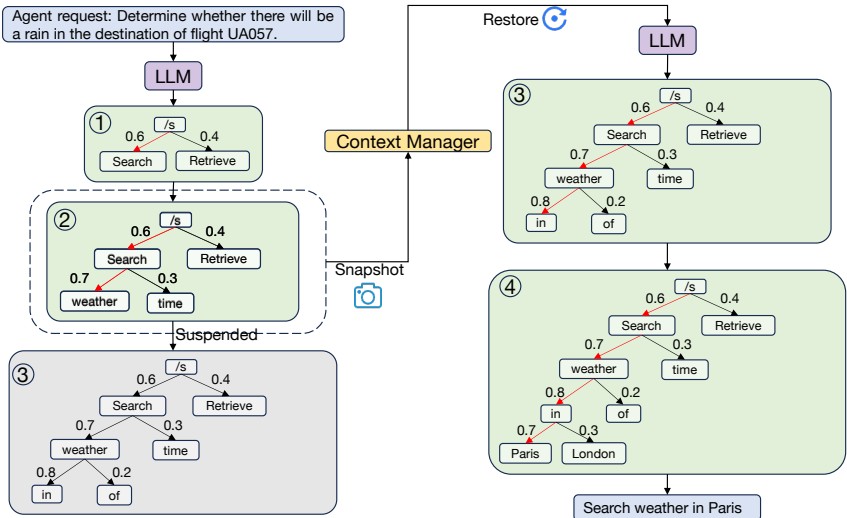

Figure 4: Illustration of the logits-based context snapshot and restoration process. We use beam search algorithm where beam width is set to 1 as an example.

*will be rain in the destination of flight UA057*, the LLM evaluates candidate tokens at each step. If suspended by the scheduler mid-generation, the context manager snapshots the intermediate results. Upon resumption, it reloads this snapshot to continue from the suspension point, reaching the final answer: *Search weather in Paris* without restarting computation.

### 3.5 Memory Manager

Unlike traditional OS memory managers handling physical RAM, AIOS's memory manager addresses agent interaction histories during runtime (Lerman & Galstyan, 2003; Zhang et al., 2024), including conversation logs and tool-calling results. It manages memory structure, allocation, read/write operations, deletion, and updates. Agent memory resides in RAM by default, but when allocated space approaches capacity, the manager implements memory swapping between RAM and disk. When an agent's memory usage exceeds its block limit (e.g., 80% of allocation), the memory manager initiates a K-Least Recently Used (LRU-K) eviction policy, transferring items from RAM to disk via the storage manager (detailed in Section 3.6). LRU-K prioritizes retaining items in RAM that have been accessed at least K times recently, while moving less frequently accessed items to disk. This balances memory efficiency by offloading infrequently accessed data while ensuring retrievability when needed. Detailed implementations of memory manager are in A.5.

### 3.6 Storage Manager

The storage manager handles persistent data storage for agents, including files or knowledge bases that agents depend on to run and the agent memories that need to be persistently stored. During an agent's runtime, when the agent's memory usage exceeds the allocated limit, the memory manager calls the storage manager to swap the data into the disk. Specifically, the storage manager reads and writes data based on the agent ID passed from the memory manager. In addition to the memory manager, the agent itself may also request to read and write data on disk during runtime, and these agent requests are also handled by the storage manager. Specifically, the agent calls the storage API in the SDK, which is further converted into storage-related system calls and put into the storage queue by the scheduler. The storage manager then processes the requests in the queue to fulfill the agent requests. The storage manager is implemented using local files and vector database (e.g., chromadb). Implementation details of the storage manager are included in Appendix A.6.

### 3.7 Tool Manager

The tool manager in the AIOS kernel is responsible for managing a broad suite of API tools supported by the AIOS SDK.

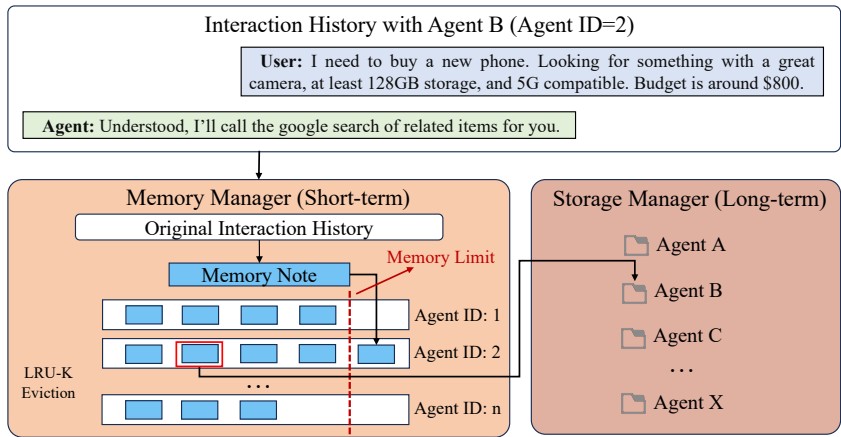

Figure 5: Illustration of memory and storage manager as well as their relationship. An agent's memory item in its memory block will be evicted to storage if its memory usage exceeds the memory limit, which is set to 80% of the memory block size. This threshold is configurable through AIOS configuration.

**Standardized Tool Loading.** The manager employs a standardized interface to handle diverse tools uniformly, while incorporating parameter validation before execution to prevent tool crashes. When invoked by name, the tool manager dynamically loads the tool instance including executables initialization and dependencies verification.

**Resolution of Tool Call Conflicts.** For tools with parallel access constraints, the system utilizes a hashmap to monitor real-time instance counts. Request processing involves hashmap verification against usage and parallel limits; upon detecting conflicts, the system advances to subsequent queue requests until identifying a conflict-free candidate. The implementation details are presented in Appendix A.7.

### 3.8 Access Manager

The access manager in the AIOS kernel provides the following two key functionalities.

**Access Control.** The access manager regulates cross-agent data read/write operations by implementing privilege-based access control mechanisms. It assigns each agent to a specific privilege group and enforces permissions through a hashmap architecture that maps agent IDs to their corresponding privilege groups. Access requests are validated against this permission structure before execution, ensuring agents can only access resources of other agents within their shared privilege domain.

**User Intervention.** To mitigate risks of irreversible operations (deletion, overwrite, privilege modification), a user intervention interface is provided for user confirmation. This mandates explicit user verification before proceeding with potentially destructive operations on files or tools. Implementation details can be found in Appendix A.8.

### 3.9 AIOS SDK

We design the AIOS SDK to streamline the development and integration of agents on the AIOS architecture. This SDK not only empowers developers to build agents that interact with the core functions in the AIOS kernel but also abstracts complex system calls, allowing developers to focus on the agent's internal workflows.

**Tool Integration.** To support diverse agent functionalities, the AIOS SDK integrates a wide range of tools sourced from various platforms and supports different cases of input-output modalities. Detailed information on these integrated tools is provided in Appendix B.3.

**Interaction Interface with the AIOS Kernel.** To facilitate the utilization of functions provided by AIOS system calls in the AIOS kernel, the SDK defines different API functions that agents can use to invoke system calls and request resources.

Table 1: Evaluation of agent performance on benchmarks w/o and w/ AIOS, respectively. Success rate (SR%) is used as the metric for all the benchmarks. "-" represents methods that failed GAIA benchmark tasks due to lack of API support.

| Method | HumanEval | MINT (Code) | GAIA | SWE-Bench-Lite |
|---|---|---|---|---|
| ReAct w/o AIOS | 48.8 | 29.4 | 5.5 | 3.9 |
| ReAct w/ AIOS | **50.6** | **30.1** | **7.3** | **4.3** |
| Reflexion w/o AIOS | 50.6 | 32.4 | 6.7 | 4.7 |
| Reflexion w/ AIOS | **51.8** | **33.8** | **7.8** | **5.1** |
| Autogen w/o AIOS | 87.8 | 42.5 | 7.3 | 4.3 |
| Autogen w/ AIOS | 87.8 | 42.5 | **9.7** | 4.3 |
| Open-Interpreter w/o AIOS | 85.4 | 45.9 | - | 4.7 |
| Open-Interpreter w/ AIOS | **86.0** | **48.7** | - | **5.1** |
| MetaGPT w/o AIOS | 82.9 | 41.1 | - | 5.9 |
| MetaGPT w/ AIOS | 82.9 | **41.8** | - | 5.9 |

**Agent Framework Adapter.** To support agents built with various agent creation frameworks, such as Autogen (Wu et al., 2023), Open-Interpreter (Lucas, 2024), and MetaGPT (Hong et al., 2023), the AIOS SDK provides adapters for these frameworks. These adapters locate the core functions in the aforementioned frameworks and redirect them to the functions in AIOS. This adaptation allows agents from different frameworks to operate within the AIOS environment without modification of the agent code. Further details on the core functions and specific adaptations for each agent framework are provided in Appendix B.5.

## 4 Evaluation

In this section, we conduct experiments to answer the following research questions.

• RQ1: Can AIOS maintain or even enhance the performance of agents on standard benchmarks when running multiple agent instances simultaneously?

• RQ2: How effectively can AIOS optimize system execution throughput and reduce response latency when serving numerous agents built with different agent frameworks?

• RQ3: How scalable is AIOS as the number of concurrently running agents increases?

### 4.1 Setup

**Models.** We use the GPT-4o-mini (Achiam et al., 2023) as the closed-source API, and use two open-source LLMs, i.e., Llama-3.1-8b (Dubey et al., 2024) and Mistral-7b (Jiang et al., 2023), as the LLM core, respectively, during the experiments. The open-source models are both instruction-tuned versions and we use float16 precision.

**Hardware.** Our experiments are conducted on an Ubuntu 22.04 machine equipped with NVIDIA RTX A5000 GPUs (24GB). We run all experiments using a single A5000 GPU.

**Agent Frameworks.** We conduct evaluation by running agents built from various popular agent frameworks: ReAct (Yao et al., 2023), Reflexion (Shinn et al., 2023), Autogen (Wu et al., 2023), Open-Interpreter (Lucas, 2024) and MetaGPT (Hong et al., 2023). Details of these agent frameworks are introduced in Appendix B.5.

**Workloads.** We evaluate on a resource-constrained scenario in which agents run concurrently with a single LLM deployed that can process only one prompt request at a time. To create these concurrent conditions, we set the maximum number of working threads to 250 by default, i.e., at most 250 agents can run concurrently at the same time. The impact of increasing the number of agents will be analyzed in Section 4.4. By default, we use RR as the scheduling strategy for AIOS to run agents. The impact of using other strategy (i.e., FIFO) is reported in Appendix D.

### 4.2 Agent Performance (RQ1)

To evaluate the impact of using AIOS on agent performance in standard benchmarks, we adopt four agent benchmarks, i.e., HumanEval (Chen et al., 2021a), MINT (the code subset) (Wang et al., 2023b), GAIA (Mialon et al., 2023) and SWE-Bench-Lite (Jimenez et al., 2024) to run agents without and with AIOS, respectively. We use the success rate

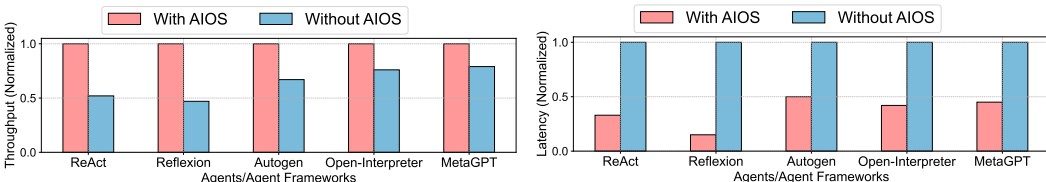

(a) Normalized throughput. Higher is better.   (b) Normalized latency. Lower is better.

Figure 6: Efficiency analysis on different agent frameworks evaluated on the Llama-3.1-8b model on the HumanEval benchmark.

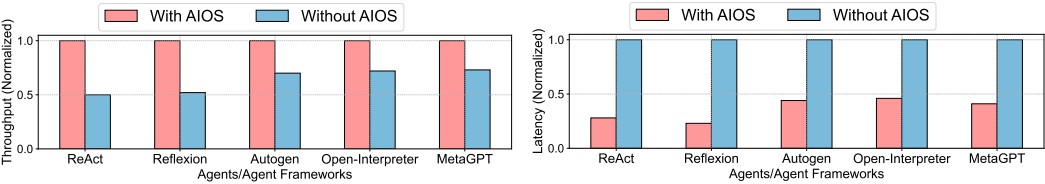

(a) Normalized throughput. Higher is better.   (b) Normalized latency. Lower is better.

Figure 7: Efficiency analysis on different agent frameworks evaluated on the Mistral-7b model on the HumanEval benchmark.

(SR%) as the metric, consistent with the original benchmarks and use GPT-4o-mini as the LLM core to run all the agents. We set the temperature as 1.0 for GPT-4o-mini in all experiments. Detailed descriptions of the benchmark setups and configurations can be found in Appendix C. As shown in Table 1, incorporating AIOS consistently maintains agent performance across standard benchmarks. In some cases, AIOS can also contribute to agent performance improvements. For example, in code generation benchmarks such as MINT, HumanEval, and SWE-Bench-Lite, AIOS boosts agent performance by prompt enhancement, which embeds the system prompts with more structural input and output within the LLM wrapper. These enhanced prompts provide the LLM with additional context and structural guidance for response generation. In tool calling benchmarks like GAIA, agent performance is even boosted by two following mechanisms: pre-execution parameter validation via structural regex to check the format of tool calls before execution, and (2) conflict resolution hashmaps to mitigate concurrent access issues.

### 4.3 Efficiency Analysis (RQ2)

In our efficiency experiments, we evaluate system performance using two key metrics: **throughput** and **latency**. Throughput is measured by counting the number of AIOS system calls executed per second, indicating the system's capacity to handle multiple requests in parallel. Latency, on the other hand, is measured as the average waiting time experienced by agents, from the moment a query is submitted to the completion of the response, reflecting the system's responsiveness. To ensure a controlled and consistent testing environment, we conduct these evaluations using the two open-source models, Llama-3.1-8b and Mistral-7b, both hosted locally. Hosting these models locally reduces potential variability in LLM API response times due to network-related latency issues. As shown in Figure 6a and Figure 7a, the results demonstrate that AIOS achieves significantly higher throughput across different agent frameworks, to a 2.1× increase in throughput when using Reflexion-based agents on Llama-3.1-8b. This improvement is attributed to the scheduling employed in the AIOS kernel, which prevents unnecessary trial-and-error attempts by avoiding prompts that cannot be loaded onto the GPU for execution. In terms of latency, as illustrated in Figure 6b and Figure 7b, the average waiting time for agents is also substantially reduced. This reduction highlights the efficiency of AIOS in serving LLM-based agents.

### 4.4 Scalability Analysis (RQ3)

We evaluated AIOS scalability by incrementally increasing active agents from 250 to 2000, using Llama-3.1-8b and Mistral-7b models on the HumanEval benchmark. We duplicated HumanEval's 164 samples to match agent counts, enabling large-scale concurrent execution. As demonstrated in Figure 8, AIOS maintains approximately linear relationships between both overall execution time and average agent waiting time relative to agent count. This

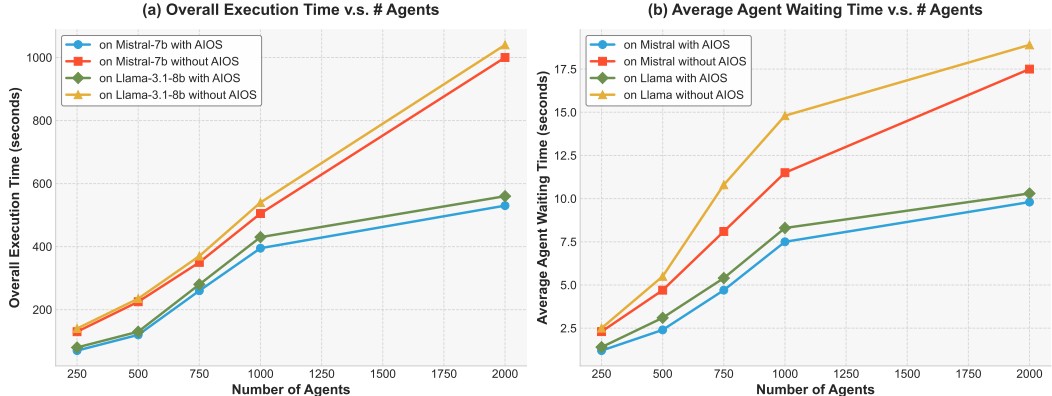

Figure 8: Overall execution time and average agent waiting time when agent number increases from 250 to 2000.

indicates that AIOS can efficiently handle workload even under increasing demand. In contrast, the gap of execution and waiting times between without AIOS and using AIOS widens as agent counts increase. This growing differential underscores AIOS's scalability under high concurrent workloads.

## 5 Related Work

**The evolution of operating systems (OS)** has progressed from rudimentary to sophisticated interactive systems. This evolution transitioned from basic batch processing (IBM, 2010) to advanced process management including time-sharing (Ritchie & Thompson, 1974) and multi-tasking (Hoare, 1974; Engler et al., 1995), enabling complex task handling. Development advanced toward modular architecture with specialized components for process scheduling (Liu & Layland, 1973; Dijkstra, 2002), memory management (Denning, 1968; Daley & Dennis, 1968), and filesystem operations (Rosenblum & Ousterhout, 1992; McKusick et al., 1984), improving system efficiency. The introduction of graphical user interfaces (GUIs) in Macintosh, Windows, and GNOME enhanced user interaction. Currently, AI models, particularly LLMs, are migrating from application to system layers, providing standardized services across applications.

**Large Language Model Agents** are used to solve complex planning and reasoning tasks (Xie et al., 2024; Ge et al., 2023a). Single agents engage with either digital environment or physical environment, which may invoke APIs (Ge et al., 2023a; Schick et al., 2023; Yao & Narasimhan, 2023; Parisi et al., 2022; Tang et al., 2023; Xie et al., 2024), browse websites (Nakano et al., 2022; Deng et al., 2023; Wu et al., 2024), or execute codes (Zhang et al., 2023; Yang et al.), while agents in the physical environment may manipulate objects (Brohan et al., 2023; Fan et al., 2022; Wang et al., 2023a), carry out lab experiments (Boiko et al., 2023; Bran et al., 2023), or make actionable decisions (Huang et al., 2022; Xiang et al., 2023). LLM-based multi-agent systems (MAS) leverage the interaction among multiple agents for problem solving. The relationship among the multiple agents could be cooperative (Wang et al., 2023c; Mandi et al., 2023), competitive (Chan et al., 2023; Du et al., 2023), or a mixture of cooperation and competition (Ge et al., 2023b). In cooperative multi-agent systems, each agent takes and assesses the information provided by other agents, thereby working together to solve complex tasks, such as role playing (Li et al., 2023; Chen et al., 2023; Zhu et al., 2023), social simulation (Park et al., 2023) and software development (Hong et al., 2023; Qian et al., 2023; Wu et al., 2023; Josifoski et al., 2023).

## 6 Conclusion and Future Work

This paper introduces AIOS, an architecture serving LLM-based agents through an innovative kernel that isolates resources and LLM services from agent applications. The complementary AIOS SDK enables agent applications to leverage kernel functionalities efficiently. Experimental validation confirms AIOS maintains or enhances agent performance on standard benchmarks while significantly accelerating execution time, improving

system throughput, and demonstrating scalability with increasing concurrent agent loads. We envision this work catalyzing future innovations that refine and expand the architecture to address evolving requirements for developing and deploying LLM-based agents.

## 7 Acknowledgement

We thank Balaji Rama, Hang Gao, Shuhang Lin, Jian Zhang and Zhenting Wang for their valuable discussions and suggestions during the project.

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

## APPENDIX

This appendix contains additional details for this paper. The appendix is organized as: Section §A provides **AIOS Kernel Implementation Details**. Section §B reports more about **AIOS SDK**. Section §C reports more **Details of Agent Benchmarks**. Section §D shows more **Additional Experimental Results**. Section §E analyzes **Discussion**.

## A  AIOS Kernel Implementation Details

### A.1  AIOS System Call

The modules in AIOS achieve their functionalities by invoking system calls. Table 2 shows a more comprehensive list of system calls correspondent to different modules and present the arguments for invoking these system calls.

Table 2: AIOS modules and their correspondent system calls.

| Module | System Call |
| --- | --- |
| LLM Core | execute_llm_syscall, get_model_response, process_model_response |
| Scheduler | execute_syscall, start, stop |
| Context Manager | generate_response_with_interruption, load_context, clear_context |
| Memory Manager | execute_memory_syscall, add_memory, remove_memory update_memory, retrieve_memory |
| Storage Manager | execute_storage_syscall, sto_create_file, sto_create_directory sto_mount, sto_write, sto_retrieve, sto_rollback, sto_share |
| Tool Manager | execute_tool_syscall, load_tool_instance |
| Access Manager | add_privilege, check_access, ask_permission |

**Thread Binding.** Each system call within AIOS is bound to a separate thread for execution, allowing for concurrent processing. The thread binding is implemented by inheriting the **Thread** class and overwrites its init and run methods.

```python
class SysCall(Thread):
    def __init__(self, agent_name, request_data):
        super().__init__()
        self.agent_name = agent_name
        self.request_data = request_data
        self.event = threading.Event()
        self.pid = None
        self.status = None
        self.response = None
        self.time_limit = None
        self.created_time = None
        self.start_time = None
        self.end_time = None

    def run(self):
        self.set_pid(self.native_id)
        self.event.wait()
```

### A.2  LLM Core

AIOS implements a unified interface through the **LLMAdapter** class, which provides a consistent function interface for integrating LLM instances from various backends. This architecture allows for seamless interaction with different LLM providers while maintaining a standardized API. Table 3 presents the supported LLM backends and their respective functionalities within the AIOS framework.

Table 3: Adapted LLM backends in AIOS and supported features.

| Backend | Supported Features in AIOS | |
| --- | --- | --- |
| | Structured Output | Function Calling |
| OpenAI (cloud) | ✓ | ✓ |
| Anthropic (cloud) | ✓ | ✓ |
| Google (cloud) | ✓ | ✓ |
| Groq (cloud) | ✓ | ✓ |
| Bedrock (cloud) | ✓ | ✓ |
| Huggingface (local) | ✓ | ✓ |
| vllm (local) | ✓ | ✓ |
| Ollama (local) | ✓ | ✓ |

```python
class LLMAdapter:
    # The LLMAdapter class is an abstraction layer wraps LLM core instances from different LLM
        backends.

    def __init__(
        self,
        llm_configs: List[Dict[str, Any]],
        api_key: Optional[Union[str, List[str]]] = None,
        log_mode: str = "console",
        use_context_manager: bool = False,
        strategy: Optional[RouterStrategy] = RouterStrategy.Sequential,
    ):
        # Initialize the LLMAdapter.

    def setup_api_keys(self) -> None:
        # Set up API keys for different providers from config or environment.
        pass

    def initialize_llms(self) -> None:
        # Initialize LLM backends based on configurations.
        pass

    def initialize_single_llm(self, config: LLMConfig) -> None:
        # Initialize a single LLM based on its configuration.
        pass

    def handle_completion_error(self, error: Exception) -> LLMResponse:
        # Handle errors that occur during LLM completion.
        pass

    def execute_llm_syscall(
        self,
        llm_syscall,
        temperature: float = 0.0
    ) -> LLMResponse:
        # Address request sent from the agent.
        pass

    def get_model_response(
        self,
        model_name: str,
        model: Union[str, HfLocalBackend, OpenAI],
        messages: List[Dict],
        tools: Optional[List],
        llm_syscall,
        api_base: Optional[str] = None,
        message_return_type: Optional[str] = "text",
        response_format: Optional[Dict[str, Dict]] = None,
        temperature: float = 1.0,
        max_tokens: int = 1000
    ) -> Any:
        # Get response from the model.

    def process_response(
        self,
        completed_response: str | List, # either a response message of a string or a list of
            tool calls
        finished: bool,
        tools: Optional[List] = None,
        model: Union[str, OpenAI, HfLocalBackend] = None,
        message_return_type: Optional[str] = None
    ) -> LLMResponse:
        # Process the model's response into the appropriate format.
        pass
```

### A.3 Scheduler

AIOS implements a flexible scheduler architecture through the **BaseScheduler** class, which serves as the foundation for various scheduling strategies. By designing an extensible inheritance model, specialized schedulers (implementing algorithms such as FIFO, Round Robin, and priority-based scheduling) can be derived from this base class. This modular design ensures that new scheduling algorithms can be introduced without modifying existing implementations, thus maintaining strong isolation between different scheduling strategies and enhancing the overall system flexibility.

```python
class BaseScheduler:
    # Task scheduler implementation.
    def __init__(
        self,
        llm: LLMAdapter,
        memory_manager: MemoryManager,
        storage_manager: StorageManager,
        tool_manager: ToolManager,
        log_mode: str,
        get_llm_syscall: LLMRequestQueueGetMessage,
        get_memory_syscall: MemoryRequestQueueGetMessage,
        get_storage_syscall: StorageRequestQueueGetMessage,
        get_tool_syscall: ToolRequestQueueGetMessage,
    ):
        # Initialize the Scheduler.
        pass

    def _execute_syscall(
        self,
        syscall: Any,
        executor: Any,
        syscall_type: str
    ) -> Optional[Dict[str, Any]]:
        # Execute a system call with proper status tracking and error handling.
        pass

    def process_llm_requests(self) -> None:
        # Process LLM requests from the queue.
        pass

    def process_memory_requests(self) -> None:
        # Process Memory requests from the queue.
        pass

    def process_storage_requests(self) -> None:
        # Process Storage requests from the queue.
        pass

    def process_tool_requests(self) -> None:
        # Process Tool requests from the queue.
        pass

    def start(self) -> None:
        # Start all request processing threads.
        pass

    def stop(self) -> None:
        # Stop all request processing threads.
        pass
```

### A.4 Context Manager

The **SimpleContextManager** implements an efficient mechanism for managing LLM generation contexts with time-aware interruption capabilities. This component enables preemptive multitasking by preserving LLM states across generation phases. When an LLM successfully begins token generation within its allocated time slice (evidenced by at least one decoded token), the context manager captures and preserves the intermediate generation state. This ensures that subsequent resumption can continue from the exact point of interruption, eliminating redundant computation. This architecture supports various LLM backends (OpenAI, HuggingFace, and others) through a unified interface while enforcing strict time boundaries on generation tasks. By managing both streaming responses and enforcing time limits, the context manager helps improve fairness among different queries to LLMs.

```python
class SimpleContextManager(BaseContextManager):
    """
    A simple context manager for handling LLM context saving and loading.

    This class provides functionality to save the current state of an LLM generation,
    load previously saved states, and manage context for different processes.
    """
    def __init__(self):
        # Initialize the SimpleContextManager with an empty context dictionary.
        pass

    def get_streaming_completion_response(
            self,
            model_or_client: Union[str, OpenAI],
            model_name: str,
            messages: List[Dict[str, str]],
            tools: Optional[List[Dict[str, Any]]],
            temperature: float,
            max_tokens: int,
            response_format: Optional[Dict[str, Any]] = None,
            stream: bool = True
    ) -> Any:
        # Get a completion response from either litellm or OpenAI client.
        pass

    def process_completion_streaming_response(
            self,
            response: Any,
            initial_content: str,
            time_limit: float
    ) -> Tuple[str, bool]:
        # Process a streaming response with time limit enforcement.
        pass

    def _is_huggingface_model(self, model) -> bool:
        # Check if the model is a HuggingFace model instance.
        pass

    def generate_with_time_limit_hf(
            self,
            model,
            messages: List[Dict[str, str]],
            max_tokens: int,
            temperature: float,
            pid: int,
            time_limit: float
    ) -> Tuple[str, bool, Dict]:
        # Generate text with a HuggingFace model with time limit enforcement.
        pass

    def generate_response_with_interruption(self,
            model_name: str,
            model: Union[str, OpenAI, Any],
            messages: List[Dict[str, str]],
            tools: Optional[List[Dict[str, Any]]],
            message_return_type: str,
            temperature: float,
            max_tokens: int,
            pid: Union[int, str],
            time_limit: float,
            response_format: Optional[Dict[str, Any]] = None
    ) -> Tuple[Any, bool]:
        # Save the context of an LLM generation. This method handles different types of LLM
        #    models (string-based, OpenAI client, or HuggingFace) and different response types
        #     (text, JSON, or tool calls). It manages streaming responses and enforces time
        #    limits.
        pass

    def load_context(self, pid):
        # Load a previously saved context for a process.
        pass

    def clear_context(self, pid):
        # Clear the saved context for a process.
        pass
```

## A.5 Memory Manager

The memory manager provides RAM-based memory operations for the AIOS system, handling transient session-specific data that clears when agent sessions end. Built on the BaseMemoryManager class, it offers comprehensive memory access through atomic operations, automatic metadata synchronization, and thread-safe access patterns. The system efficiently processes memory through a complete suite of CRUD operations and advanced retrieval mechanisms, functioning similarly to pointer management in low-level programming while ensuring data integrity during concurrent operations.

```python
class BaseMemoryManager:
    # This class provides the core functionality for memory operations in the AIOS system,
    #     including adding, removing, updating, and retrieving memories. It acts as a wrapper
    #     for memory access, similar to working with pointers in low-level languages.

    def __init__(self, log_mode):
        # Initialize the BaseMemoryManager.
        pass

    def _analyze_query_to_memory(self, query: MemoryQuery) -> 'MemoryNote':
        # Convert a MemoryQuery to a MemoryNote object.
        pass

    def execute_memory_syscall(self, memory_syscall):
        # Route a memory syscall to the appropriate method.
        pass

    def add_memory(
            self,
            memory_note
    ):
        # Add a memory note to the storage.
        pass

    def remove_memory(
            self,
            memory_id
    ):
        # Remove a memory note from storage.
        pass

    def update_memory(
            self,
            memory_note
    ):
        # Update an existing memory note.
        pass

    def get_memory(
            self,
            memory_id: str
    ) -> 'MemoryNote':
        # Retrieve a memory note by ID.
        pass

    def _retrieve_memory_raw(
            self,
            memory_query: MemoryQuery
    ):
        # Retrieve memories similar to the query content.
        pass

    def retrieve_memory(
            self,
            memory_query: MemoryQuery
    ):
        # Retrieve memories similar to the query content.
        pass
```

## A.6 Storage Manager

The Storage Manager orchestrates persistent data operations in AIOS, combining traditional file storage with vector database capabilities. It implements versioned file management, thread-safe access through file-specific locks, and semantic search functionality. The storage

manager provides file operations including file management, semantic file retrieval, version control with rollback capabilities, and file sharing. Function interfaces are shown as below.

```python
class StorageManager:
    """
    Storage manager provides versioning, locking, and vector database integration for
        efficient file operations and retrieval.
    """
    def __init__(self, root_dir, use_vector_db=True, max_versions=20):
        # Initialize LSFS with the specified root directory and configuration.
        pass

    def __del__(self):
        # Destructor that stops the file system observer when the LSFS instance is deleted.
        pass

    def get_file_hash(self, file_path: str) -> str:
        # Generate a SHA-256 hash for a file path.
        pass

    def get_file_lock(self, file_path: str) -> threading.Lock:
        # Get or create a thread lock for a specific file path.
        pass

    def handle_file_change(self, file_path: str, change_type: str):
        # Handle file changes with proper lock management.
        pass

    def get_file_history(self, file_path: str, limit: int = None) -> list:
        # Retrieve version history for a file from Redis cache.

    def restore_version(self, file_path: str, version_index: int) -> bool:
        # Restore a file to a previous version.
        pass

    def execute_storage_syscall(self, storage_syscall):
        # Process and route storage syscalls to appropriate file system operations.
        pass

    def sto_create_file(self, file_name: str, file_path: str, collection_name: str = None) ->
         bool:
        # Create a new empty file in the file system.
        pass

    def sto_create_directory(self, dir_name: str, dir_path: str, collection_name: str = None)
         -> bool:
        # Create a new directory in the file system.
        pass

    def sto_mount(self, collection_name: str, root_dir: str) -> str:
        # Mount a directory for an agent and build the vector database.
        pass

    def sto_retrieve(
            self,
            collection_name: str,
            query_text: str,
            k: str = "3",
            keywords: str = None
        ) -> list:
        # Retrieve documents from the vector database using semantic search.

    def sto_rollback(
            self,
            file_path,
            n=1,
            time=None
        ) -> bool:
       # Roll back a file to a previous version by index or timestamp.
        pass

    def generate_share_link(self, file_path: str) -> str:
        # Generate a publicly accessible link for sharing a file.
        pass

    def sto_share(self, file_path: str, collection_name: str = None) -> dict:
        # Share a file by generating a public access link with proper lock management.
        pass
```

### A.7 Tool Manager

The tool manager module is responsible for loading tools executing tools with tool conflict prevention mechanisms. Implementation details are shown below.

```python
class ToolManager:
    def __init__(
        self,
        log_mode: str = "console",
    ):
        pass

    def execute_tool_syscall(self, tool_syscall) -> ToolResponse:
        pass

    def load_tool_instance(self, tool_org_and_name):
        pass
```

### A.8 Access Manager

The access manager provides two key functions: First is to check access when agents attempt to access other agents' resources. Second is to request user permission before agents execute irreversible actions such as deletion of files. Function interfaces are shown below.

```python
class AccessManager:
    def __init__(self):
        pass

    def add_privilege(self, sid, tid):
        # Assigns an agent into another agent's privilege group
        pass

    def check_access(self, sid, tid):
        # Checks if the source agent is in the target agent's priviledge group
        pass

    def ask_permission(self, operation):
        # Prompts the user for confirmation before an irreversible operation
        pass
```

### A.9 Module Hooks

To effectively separate the interface of calling the AIOS kernel modules from the implementation details, we employ a hook mechanism to initialize modules and export the necessary call interfaces. Here are the hooks we use for initializing modules.

```python
@validate(LLMParams)
def useLLM(params: LLMParams) -> LLM:
    """ Initialize and return a Language Learning Model (LLM) instance.

    Args:
        params (LLMParams): Parameters required for LLM initialization.

    Returns:
        LLM: An instance of the initialized LLM.
    """
    return LLM(**params.model_dump())
```

```python
@validate(MemoryManagerParams)
def useMemoryManager(params: MemoryManagerParams) -> MemoryManager:
    """ Initialize and return a memory instance.

    Args:
        params (MemoryParams): Parameters required for Memory Manager Initialization.

    Returns:
        Memory Manager: An instance of the initialized Memory Manager.
    """
    return MemoryManager(**params.model_dump())
```

```python
@validate(StorageManagerParams)
def useStorageManager(params: StorageManagerParams) -> StorageManager:
    """  Initialize and return a storage instance.

    Args:
        params (StorageManagerParams): Parameters required for Memory Manager Initialization.

    Returns:
        Storage Manager: An instance of the initialized Storage Manager.
    """
    return StorageManager(**params.model_dump())
```

```python
@validate(ToolManagerParams)
def useToolManager(params: ToolManagerParams) -> ToolManager:
    """  Initialize and return a tool instance.

    Args:
        params (ToolManagerParams): Parameters required for Tool Manager Initialization.

    Returns:
        Tool Manager: An instance of the initialized Tool Manager.
    """
    return ToolManager(**params.model_dump())
```

# B   AIOS SDK

The AIOS SDK provides a structured interface between user device applications and the AIOS kernel through different functional modules. All queries from these modules are ultimately channeled through a central **send_request()** function in the SDK, which then communicates with the AIOS kernel via HTTP requests (either to localhost or a remote URL). From the agent developer's perspective, their agents are essentially composed of code snippets which could include agent logic and resource-related commands for LLM, memory, storage and tool that interact with their respective modules through the SDK apis. This creates a clean separation between the application logic and request of kernel resources.

## B.1   Query and Response

The AIOS SDK defines a robust query-response architecture that enables seamless communication between agent applications and the AIOS kernel. This framework is built on two foundational data structures: Query and Response, which facilitate structured data exchange across the system.

**Query Structure.** The Query class serves as the abstract base for all input requests, establishing a consistent interface for agent interactions with the kernel. It branches into four specialized implementations:

- LLMQuery: Facilitates natural language interactions with configurable parameters for temperature, token limits, and various action types including chat, JSON output, tool calls, and file operations.

- MemoryQuery: Handles transient data operations including adding, retrieving, updating, and removing memories with specialized support for agentic memory management.

- StorageQuery: Manages persistent storage operations through parameterized requests for file and directory manipulation.

- ToolQuery: Enables access to external capabilities through structured tool calls, extending the system's functionality.

**Response Structure.** The Response class provides a complementary structure for standardized outputs from the AIOS kernel. Each response type corresponds to its query counterpart:

- LLMResponse: Returns generated text, tool call outcomes, completion status, and error information from language model operations.

- MemoryResponse: Delivers memory content, metadata, search results, and operation status for memory management functions.

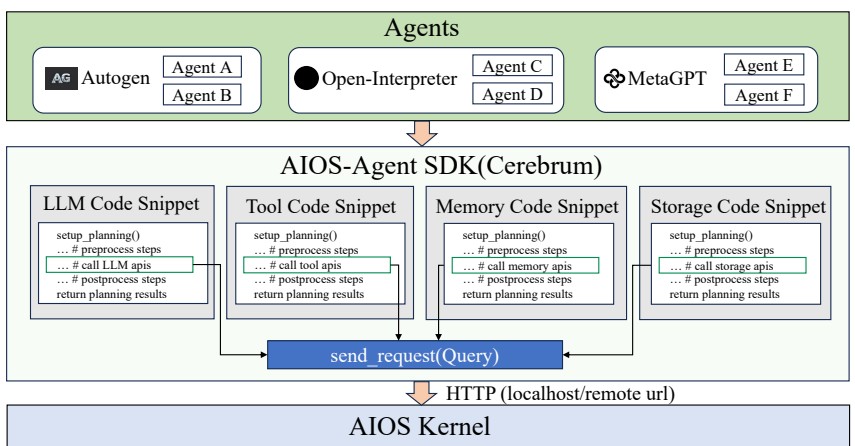

Figure 9: Illustration of how agent applications leverage the AIOS SDK to send queries to the AIOS Kernel. For simplicity, queries sent directly to the OS kernel are omitted.

- `StorageResponse`: Provides operation outcomes, completion status, and error details for storage-related activities.

- `ToolResponse`: Returns tool execution results, completion status, and error information from external tool operations.

```python
class Query(BaseModel):
    """
    Base class for all query types in the AIOS system.

    This class serves as the foundation for specialized query classes like LLMQuery,
    MemoryQuery, StorageQuery, and ToolQuery. It defines the minimum structure required
    for a valid query within the AIOS ecosystem.

    Attributes:
        query_class: Identifier for the query type, must be one of
                     ["llm", "memory", "storage", "tool"]
    """
    query_class: Literal["llm", "memory", "storage", "tool"]
```

```python
class Response(BaseModel):
    """
    Base class for all response types in the AIOS system.
    This class serves as the foundation for specialized response classes like LLMResponse,
    MemoryResponse, StorageResponse, and ToolResponse. It defines the minimum structure
    required for a valid response within the AIOS ecosystem.
    """
    response_class: Literal["llm", "memory", "storage", "tool"]
```

```python
class LLMQuery(Query):
    query_class: str = "llm"
    llms: Optional[List[Dict[str, Any]]] = Field(default=None)
    messages: List[Dict[str, Union[str, Any]]]
    tools: Optional[List[Dict[str, Any]]] = Field(default_factory=list)
    action_type: Literal["chat", "chat_with_json_output", "chat_with_tool_call_output", "
        call_tool", "operate_file"] = Field(default="chat")
    temperature: float = Field(default=1.0)
    max_new_tokens: int = Field(default=1000)
    message_return_type: Literal["text", "json"] = Field(default="text")
    response_format: Optional[Dict[str, Any]] = Field(default=None)

    class Config:
        arbitrary_types_allowed = True
```

```python
class LLMResponse(Response):
    """
    Response class for LLM operations.
    This class represents the output structure after performing LLM actions.
    """
    response_class: str = "llm"
    response_message: Optional[str] = None
    tool_calls: Optional[List[Dict[str, Any]]] = None
    finished: bool = False
    error: Optional[str] = None
    status_code: int = 200

    class Config:
        arbitrary_types_allowed = True
```

```python
class MemoryQuery(Query):
    """
    Query class for memory operations.
    """
    query_class: str = "memory"
    operation_type: Literal["add_memory", "get_memory", "update_memory", "remove_memory", "
        retrieve_memory", "add_agentic_memory","retrieve_memory_raw"]
    params: Dict[str, Any] = Field(default_factory=dict)

    class Config:
        arbitrary_types_allowed = True
```

```python
class MemoryResponse(Response):
    """
    Response class for memory operations.
    """
    response_class: str = "memory"
    memory_id: Optional[str] = None
    content: Optional[str] = None
    metadata: Optional[Dict[str, Any]] = None
    search_results: Optional[List[Dict[str, Any]]] = None
    success: bool = False
    error: Optional[str] = None
    # status_code: int = 200

    class Config:
        arbitrary_types_allowed = True
```

```python
class StorageQuery(Query):
    """
    Query class for storage operations.
    """
    query_class: str = "storage"
    params: Dict[str, Union[str, Any]]
    operation_type: str = Field(default="text")

    class Config:
        arbitrary_types_allowed = True
```

```python
class StorageResponse(Response):
    """
    Response class for storage operations.
    """
    response_class: str = "storage"
    response_message: Optional[str] = None
    finished: bool = False
    error: Optional[str] = None
    status_code: int = 200
```

```
class ToolQuery(Query):
    """
    Query class for tool operations.
    """
    query_class: str = "tool"
    tool_calls: List[Dict[str, Union[str, Any]]]

    class Config:
        arbitrary_types_allowed = True
```

```
class ToolResponse(Response):
    """
    Response class for tool operations.
    """
    response_class: str = "tool"
    response_message: Optional[str] = None
    finished: bool = False
    error: Optional[str] = None
    status_code: int = 200
```

## B.2 AIOS SDK APIs

AIOS SDK also provides multiple API functions that are constructed using the Query and Response data structure and send_request() functions. Avaiable APIs are shown in Table 4.

Table 4: AIOS SDK APIs.

| Module | APIs |
| --- | --- |
| LLM Core | llm_chat, llm_chat_with_json_output, llm_chat_with_tool_call_output llm_call_tool, llm_operate_file |
| Memory | create_memory, get_memory, delete_memory, update_memory search_memories |
| Storage | mount, retrieve_file, create_file, create_dir, write_file, rollback_file share_file |
| Tool | call_tool |

## B.3 Supported Tools.

As is illustrated in Table 5, AIOS SDK integrates diverse computational tools to address a wide spectrum of information processing tasks. The SDK incorporates 17 native tools spanning multiple modalities and functionalities, enabling sophisticated interaction patterns across text, image, and audio domains. These tools can be categorized into three primary sources: established technology providers (Google, Bing, WolframAlpha), specialized API hubs (Rapid API Hub), and advanced AI model providers (Huggingface).

The toolkit's architecture demonstrates particular strength in text-based operations, with 12 tools supporting text input or output modalities. This includes fundamental information retrieval services (Arxiv, BingSearch, Wikipedia), specialized analytical tools (Currency-Converter, MoonPhaseSearch), and domain-specific applications (ImdbRank, TripAdvisor). Furthermore, the SDK exhibits robust cross-modal capabilities through tools like VisualQuestionAnswering (image-text integration), TextToAudio (text-to-speech synthesis), and VoiceActivityRecognition (speech-to-text conversion).

## B.4 Agent Examples

Here we provide examples of agents developed by leveraging AIOS SDK.

**Travel Agent:** The travel agent is designed for trip planning, including searching for flights, accommodations, and local activities.

**Rec Agent:** The recommendation agent is designed for suggesting movies and TV series.

**Math Agent:** This agent is designed to solve equations, perform calculations, and provide step-by-step explanations for different math problems.

Table 5: Tools supported by AIOS SDK, ordered by names in alphabet.

| Tool Name | Source | Type | Modality (Input → Output) |
|---|---|---|---|
| Arxiv | Arxiv | API | Text → Text |
| BingSearch | Bing | API | Text → Text |
| CurrencyConverter | Rapid API Hub | API | Text → Text |
| GooglePlace | Google | API | Image/Text → Text |
| GoogleSearch | Google | API | Text → Image |
| ImageCaption | Huggingface | Local Model/API | Text → Text |
| ImdbRank | Rapid API Hub | API | Text → Text |
| MoonPhaseSearch | Rapid API Hub | API | Text → Text |
| Shazam | Rapid API Hub | API | Text → Text/Audio |
| TextToAudio | Huggingface | Local Model/API | Text → Audio |
| TextToImage | Huggingface | Local Model/API | Text → Image |
| TripAdvisor | Rapid API Hub | API | Text → Text |
| VisualQuestionAnswering | Huggingface | Local Model/API | Image & Text → Text |
| VoiceActivityRecognition | Huggingface | Local Model/API | Audio → Text |
| Wikipedia | Wikipedia | API | Text → Text |
| WolframAlpha | WolframAlpha | API | Text → Text |
| WordsAPI | Rapid API Hub | API | Text → Text |

**Creation Agent:** The creation agent is tailored for content generation tasks, such as writing, graphic design, or even video editing.

**Academic Agent:** The academic agent is designed to support research and learning, such as literature reviews and provide explanations on complex academic topics.

---

**TravelAgent Profile**

**Description:** You are an expert in planning and managing travel itineraries.
**Workflow:**
1. Identify the destination and search for hotel locations using the hotel_location_search tool.
2. Based on the hotel locations, find suitable hotels using the hotel_search tool, and select the best one.
3. Get detailed information about the selected hotel using the get_hotel_details tool.
4. Search for the nearest airport to the origin using the airport_search tool.
5. Search for the nearest airport to the destination using the airport_search tool.
6. Find available flights to the destination airport using the flight_search tool using the correct date.
7. Search for restaurant locations near destination using the restaurant_location_search tool.
8. Based on the restaurant locations, find suitable restaurants using the restaurant_search tool.
9. Get detailed information about the selected restaurants using the get_restaurant_details tool.
10. Gather additional relevant information about the destination the user is visiting using the wikipedia tool.
11. Integrate the information gathered from the previous steps to provide a comprehensive travel plan.

**Available tools:**
1. TripAdvisor
2. Wikipedia

**Example of task inputs:** I want to take a trip to Paris, France from July 4th to July 10th, 2024, and I am traveling from New York City. Help me plan this trip.

---

**RecAgent Profile**

**Description:** You are an expert who is good at recommending TV series and movies.
**Workflow:**
1. Identify the tool that you need to call to obtain information.
2. Based on the information, give recommendations for the user based on the constrains.

**Available tools**:
1. TripAdvisor
2. Wikipedia

**Example of task inputs:** Recommend three action movies from the past five years ranked between 1 and 20 with ratings above 8.0.

---

**CreationAgent Profile**

**Description:** You are an expert who is good at content creation.
**Workflow:**
1. Convert the vague description of the content requirements into concrete objects and fill in more details.
2. Identify the tool to call the tool to create content based on the filled details.

**Available tools:**
1. SDXL-Turbo

**Example of task inputs:** Create an image of a sleek, high-tech futuristic city with a vibrant nightlife atmosphere.

---

**MathAgent Profile**

**Description:** You are an expert who is good at solving mathematical problems.

**Workflow:**
1. Identify the tool to call to do some pre-calculation.
2. Perform mathematical operations using the pre-calculated result, which could involve addition, subtraction, multiplication, or division with other numeric values to solve the problem.

**Available tools:**
1. Currency Converter
2. WolframAlpha

**Example of task inputs:** Convert 15000 MXN to Canadian Dollars and find out how much it would be in USD if 1 CAD equals 0.79 USD.

---

**AcademicAgent Profile**

**Description:** You are an expert who is good at looking up and obtaining information from academic articles.

**Workflow:**
1. Identify the tool to call based on the academic requirements and call the tool.
2. Gather the information obtained from the tool to write an outline or summarization.

**Available tools:**
1. Arxiv API

**Example of task inputs:** Summarize recent studies on the role of artificial intelligence in drug discovery from 2018 to 2023.

### B.5 Support of Agent Frameworks

The core idea to adapt agents built by existing agent frameworks for AIOS is to identify the core functions that will interact with system resources and change that with functions in our native adapters. In this section, we illustrate the crucial adaptation function that needs to be changed to run agents built by other agent frameworks on AIOS.

**ReAct (Yao et al., 2023).** The ReAct framework integrates reasoning and action steps in language models, allowing them to generate intermediate reasoning traces alongside actionable steps for complex task completion. This dual approach helps models not only plan and track their thought process but also interact with external tools, improving performance on tasks like question answering, game environments, and decision-making problems that require multi-step reasoning and adaptability. By alternating between reasoning and action, ReAct reduces errors from solely predictive responses and enables more accurate, contextually aware task completion.

**Reflexion (Shinn et al., 2023).** The Reflexion framework enhances language agents with a feedback-driven mechanism, allowing them to learn from mistakes and adapt behavior through self-reflective feedback loops. By leveraging verbal reinforcement learning, agents assess and adjust their actions, which improves performance on complex tasks through iterative learning. This approach makes language agents more resilient and adaptive, enabling them to handle tasks with evolving requirements and uncertainty.

**Autogen (Wu et al., 2023).** AutoGen introduces a framework that leverages multiple language model agents with distinct roles (such as Planner, Executor, and Reflector) to collaboratively solve complex tasks through structured, goal-oriented conversations. By enabling agents to communicate and share intermediate results, AutoGen coordinates multi-step processes like data analysis, decision-making, and iterative problem-solving, significantly enhancing efficiency and accuracy beyond a single model's capabilities. This approach empowers next-generation applications, allowing LLMs to tackle dynamic workflows, adapt to task-specific nuances, and achieve higher performance in real-world scenarios. Below is the code of adapting Autogen for AIOS. Due to ongoing refactoring work by the Autogen team, only Autogen-0.2 (the latest stable version) is supported.

```python
@add_framework_adapter("AutoGen~0.2")
def prepare_autogen_0_2():
    """
    Replace OpenAIWrapper and ConversableAgent methods with aios's implementation.

    This function is used to adapt autogen's API to aios's API, and it is used
    internally by aios.
    """
    # Replace OpenAIWrapper method
    OpenAIWrapper.__init__ = adapter_autogen_client_init
    OpenAIWrapper.create = adapter_client_create
    OpenAIWrapper.extract_text_or_completion_object =
        adapter_client_extract_text_or_completion_object

    # Replace agent method
    ConversableAgent._print_received_message = _adapter_print_received_message
    ConversableAgent._generate_oai_reply_from_client = _adapter_generate_oai_reply_from_client
    ConversableAgent.generate_tool_calls_reply = adapter_generate_tool_calls_reply
    ConversableAgent.execute_function = adapter_execute_function
    ConversableAgent._a_execute_tool_call = _adapter_a_execute_tool_call
    ConversableAgent.update_tool_signature = adapter_update_tool_signature
    ConversableAgent.__init__ = adapter_autogen_agent_init
```

**Open-Interpreter (Lucas, 2024).** Open Interpreter is an open-source framework that enables users to interact with LLMs through a ChatGPT-like interface to interpret and execute complex instructions across programming languages directly in the terminal. It supports both locally-hosted and cloud-based LLMs, allowing for streamlined code execution and debugging in natural language. By translating natural language instructions into executable code, Open Interpreter offers an intuitive environment that not only simplifies development workflows but also facilitates learning by providing detailed explanations and interactive support for various coding challenges, making it suitable for developers at all skill levels. Below is the core function to be adapted for Open-Interpreter.

```python
@add_framework_adapter("Open-Interpreter")
def prepare_interpreter():
    # Prepare the interpreter for running LLM in aios.

    # Set the completion function in the interpreter
    interpreter.llm.completions = adapter_aios_completions

def adapter_aios_completions(**params):
    # aios completions replace fixed_litellm_completions in interpreter
    # Run completion
    attempts = 2
    first_error = None

    for attempt in range(attempts):
        try:
            send_request = get_request_func()
            response = send_request(
                query=LLMQuery(
                    messages=params['messages'],
                    tools=(params["tools"] if "tools" in params else None)
                )
            )["response"]

            # format similar to completion in interpreter
            comletion = {'choices':[{'delta': {}}]}
            comletion["choices"][0]["delta"]["content"] = response["response_message"]
            if response.tool_calls is not None:
                comletion["choices"][0]["delta"]["tool_calls"] =
                    format_tool_calls_to_interpreter(response["tool_calls"])

            return [comletion]  # If the completion is successful, exit the function
        except KeyboardInterrupt:
            print("Exiting...")
            sys.exit(0)
        except Exception as e:
            if attempt == 0:
                # Store the first error
                first_error = e

    if first_error is not None:
        raise first_error
```

**MetaGPT (Hong et al., 2023).** MetaGPT proposes a meta-programming approach that optimizes LLM-driven multi-agent systems by integrating task-oriented programming paradigms for complex, collaborative problem-solving. MetaGPT encodes Standardized Operating Procedures (SOPs) directly into structured prompt sequences, creating streamlined workflows that empower agents with human-like domain expertise to systematically verify intermediate outputs and proactively mitigate errors. Along this line, MetaGPT addresses the limitations of existing LLM-based frameworks, such as hallucination and cascading errors during agent chaining. This framework facilitates the decomposition of complex tasks into manageable, interdependent subtasks, improving overall system robustness, especially in high-stakes, iterative processes where reliability across agent interactions is crucial. Below is the core function to be adapted for MetaGPT.

```python
@add_framework_adapter("MetaGPT")
def prepare_metagpt():
    """
    Prepare the metagpt module to run on aios.
    """
    prepare_metagpt_config()

    BaseLLM.aask = adapter_aask
    async def adapter_aask(
        self,
        msg: Union[str, list[dict[str, str]]],
        system_msgs: Optional[list[str]] = None,
        format_msgs: Optional[list[dict[str, str]]] = None,
        images: Optional[Union[str, list[str]]] = None,
        timeout=USE_CONFIG_TIMEOUT,
        stream=True,
    ) -> str:
        rsp = await adapter_acompletion_text(message, stream=stream, timeout=self.get_timeout(
            timeout))
        return rsp if rsp else ""
```

## C  Details of Agent Benchmarks

### C.1  HumanEval

The authors (Chen et al., 2021b) [1] introduced HumanEval, a benchmark dataset comprising 164 handwritten programming problems for evaluating functional correctness of code generation models. Each problem consists of a function signature, docstring, implementation body, and comprehensive test suite, with an average of 7.7 test cases per problem. The hand-written nature of these problems is crucial, given that modern language models are typically trained on large portions of GitHub code containing existing solutions to programming challenges and contest problems. HumanEval is designed to assess multiple aspects of code generation capability: natural language comprehension, logical reasoning, algorithmic thinking, and mathematical operations. Through this publicly available benchmark, researchers can conduct rigorous and standardized evaluations of code generation models.

### C.2  MINT

MINT (Wang et al., 2023b)[2] introduced a benchmark to evaluate LLMs' ability to solve challenge tasks through multi-turn interactions. The benchmark focuses on code generation, decision making, and reasoning tasks that require LLMs to utilize tools and incorporate natural language feedback. MINT was constructed by curating multiple single-turn datasets, reducing an original collection of 29,307 instances to 586 carefully selected examples. The benchmark uses success rate (SR) as its primary evaluation metric, measuring the percentage of successfully completed tasks. For a given interaction limit $k$ ranging from 1 to 5, each LLM is allowed up to $k$ turns of interaction, with performance measured as $SR_k$. In our experiments, we set $k = 5$ and focus exclusively on MINT's code generation subset.

### C.3  GAIA

General AI Assistant (GAIA) (Mialon et al., 2023)[3] is a benchmark designed to represent a significant milestone in AI research by evaluating fundamental capabilities essential for general intelligence. Unlike traditional benchmarks that focus on specialized professional knowledge, GAIA emphasizes everyday tasks that require core abilities including logical reasoning, multi-modal processing, web navigation, and effective tool utilization. GAIA comprises 466 questions that evaluate AI assistants across multiple capabilities including reasoning, multi-modal understanding, coding, and tool usage (particularly web browsing), with tasks involving various data formats like PDFs, spreadsheets, images, videos, and audio. The benchmark organizes questions into three difficulty levels based on the number of required steps and tools: Level 1 requires minimal tool usage ($\leq 5$ steps), Level 2 demands multiple tools and 5-10 steps, while Level 3 tests advanced general assistance capabilities through complex, multi-step sequences requiring diverse tool combinations. Additionally, while web browsing is central to GAIA, the benchmark deliberately excludes complex web interactions like file uploads or posting comments, leaving such evaluations for future research.

### C.4  SWEBench-Lite

SWE-bench (Jimenez et al., 2024)[4] is a software engineering benchmark constructed through a rigorous three-stage pipeline that processes GitHub pull requests (PRs) from 12 popular Python repositories. The pipeline filters approximately 90,000 PRs based on attributes (issue resolution and test contribution) and execution criteria (successful installation and fail-to-pass test transitions), resulting in 2,294 high-quality task instances. Each task requires models to generate patch files that resolve software issues, with success determined by comprehensive test coverage. The benchmark distinguishes itself through real-world challenges, extensive input context (averaging 195 words per issue), cross-context editing requirements (typically spanning 1.7 files and 32.8 lines per solution), and robust test-based evaluation. Notably, SWE-bench's automated collection process enables continuous updates with new task instances from GitHub repositories, ensuring benchmark relevance over time.

---

[1]The dataset can be found at `https://www.github.com/openai/human-eval`.
[2]`https://xwang.dev/mint-bench/`
[3]`https://huggingface.co/gaia-benchmark`
[4]`https://www.swebench.com/`

Table 6: Impact of using different scheduling strategies, where NONE represents without using AIOS, FIFO and RR represent using AIOS with the two different scheduling strategies. All metrics are reported in seconds.

| Strategy | Overall execution time | Agent waiting time | |
|---|---|---|---|
| | | Avg. | p90 |
| None | 152.1 | 9.8 | 11.0 |
| FIFO | **74.2** | **3.0** | 5.0 |
| RR | 77.3 | 3.2 | **4.2** |

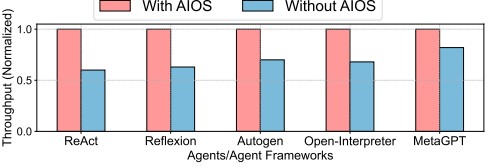

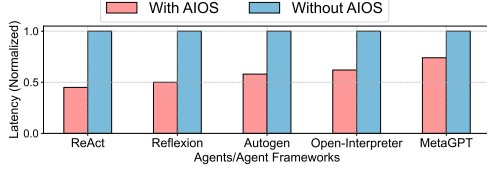

(a) Normalized throughput. Higher is better.  (b) Normalized latency. Lower is better.

Figure 10: Efficiency analysis on different agent frameworks evaluated on the Llama-3.1-8b model on the MINT benchmark.

# D  Additional Experimental Results

## D.1  Efficiency analysis.

We also report the throughput and latency of running agents on other three benchmarks on Llama-3.1-8b and Mistral-7b compared between using AIOS and without using AIOS. The results are shown in Figure 10 and Figure 11, Figure 12 and Figure 13, Figure 14 and Figure 15, respectively.

**Impact of Different Scheduling Strategies.**   To further analyze the impact of different scheduling strategies on system efficiency, we conduct an ablation study using agents built with ReAct on the HumanEval benchmark with the Llama-3.1-8b model. We test three strategies: without AIOS, FIFO, and Round Robin (RR), and measure the overall execution time and agent waiting time (average and p90).

As shown in Table 6, the FIFO strategy achieves the shortest overall execution time compared to the other strategies. RR comes second in terms of overall execution and average agent waiting time, as its context switching introduces additional overhead. However, RR performs better on the p90 metric (i.e., the value below which 90% of waiting times fall) due to its fairer scheduling approach, which reduces the likelihood of later tasks having longer waiting time, which can typically occur in FIFO.

## D.2  Correctness of context switch.

To assess the correctness of the context switch supported by the context manager, we employ the BLEU score (Papineni et al., 2002) and BERT score (Zhang et al., 2019) to measure text similarity. The similarity is calculated against the final outputs generated for the same agent under the same conditions, only varying with context switch enabled and disabled. As demonstrated in Table 7, both BLEU and BERT scores achieve a value of 1.0. The suggests that the context switch does not introduce discrepancies in output quality, suggesting the reliability of the AIOS.

# E  Discussion

## E.1  Ethical Consideration

In this section, we discuss both potential positive and negative societal impacts of the work.

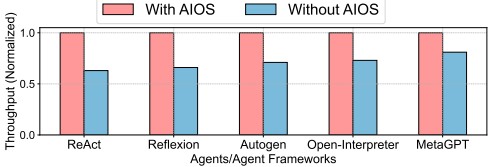
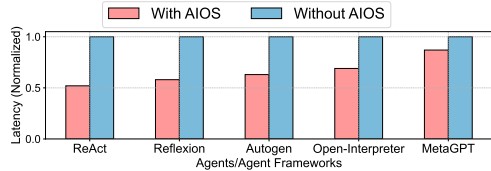

(a) Normalized throughput. Higher is better.

(b) Normalized latency. Lower is better.

Figure 11: Efficiency analysis on different agent frameworks evaluated on the Mistral-7b model on the MINT benchmark.

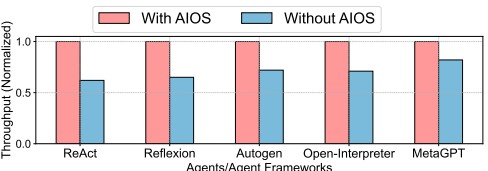
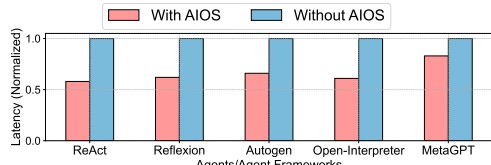

(a) Normalized throughput. Higher is better.

(b) Normalized latency. Lower is better.

Figure 12: Efficiency analysis on different agent frameworks evaluated on the Llama-3.1-8b model on the GAIA benchmark.

The potential positive societal impacts include: 1) Enhanced efficiency and productivity: AIOS can automate routine tasks, achieve more efficient operations, optimize resource allocation, and reduce bottlenecks, leading to better service and improved efficiency for agent developers; 2) Improved user experience: with better context, memory, and storage management, AIOS can offer more personalized and responsive interactions, enhancing user satisfaction across various applications; 3) Innovation ecosystem: the creation of AIOS could foster a vibrant ecosystem of agent developers and researchers, driving innovation in AI technologies and applications.

The potential negative societal impacts include: 1) Privacy concerns: the integration of LLMs into operating systems may raise privacy concerns, as AI models such as LLMs may require access to personal data to provide effective services; 2) Security risks: as AI systems become more integral to critical infrastructure, they could become targets for cyberattacks, potentially compromising sensitive data and operations; 3) System failures: the failure of integrated systems could have widespread consequences, affecting multiple sectors simultaneously and causing disruptions.

Balancing the impacts: To maximize the positive impacts and mitigate the negative ones, it is crucial to adopt a balanced approach to the development and deployment of AIOS, such as 1) Rules and standards: Implementing responsible development rules and standards to ensure data privacy, security, and ethical use of AI; 2) Robust design: implementing robust system design, regular maintenance, comprehensive testing, continuous monitoring, backup

Table 7: Correctness of context switch (text-based and logits-based), which checks the similarity between the generated final outputs with context switch enabled and disabled.

| LLM Core | Method | BLEU Score | BERT Score |
|---|---|---|---|
| Mistral-7B | Text-based | 1.0 | 1.0 |
| | Logits-based | 1.0 | 1.0 |
| Llama-3-8B | Text-based | 1.0 | 1.0 |
| | Logits-based | 1.0 | 1.0 |

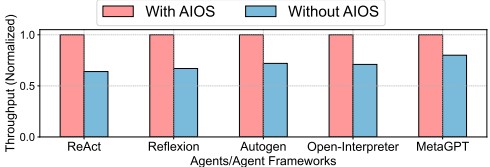 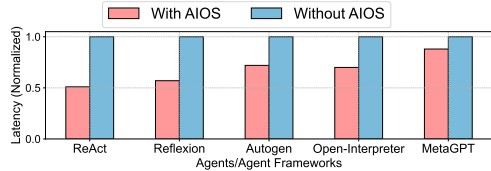

(a) Normalized throughput. Higher is better.     (b) Normalized latency. Lower is better.

Figure 13: Efficiency analysis on different agent frameworks evaluated on the Mistral-7b model on the GAIA benchmark.

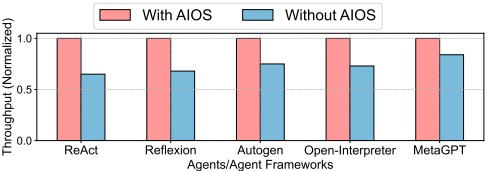 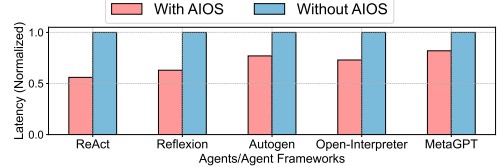

(a) Normalized throughput. Higher is better.     (b) Normalized latency. Lower is better.

Figure 14: Efficiency analysis on different agent frameworks evaluated on the Llama-3.1-8b model on the SWE-Bench-Lite benchmark.

and recovery plans, developer training, careful documentation, clear communication, and leveraging AI for predictive maintenance and automated recovery; 3) Public engagement: engaging with the public to raise awareness about the benefits and challenges of AI, ensuring that societal concerns are addressed in the development process.

By addressing these considerations, society can harness the potential of AIOS while mitigating its risks, leading to a more equitable and prosperous future.

### E.2   Future Directions

With AIOS as a foundation, there are many directions for future research to pursue. This section outlines potential areas of study that expand upon AIOS.

**Semantic Scheduling Algorithms.** The scheduling function of AIOS lays the groundwork for the development of more advanced algorithms. Future research could focus on algorithms that perform dependency analysis among agent requests, optimizing the allocation of computational resources. Additionally, some of the tool resources are locally deployed models, which can also be incorporated into the scheduling paradigm. This includes the management of tool status and snapshots, suggesting a move towards a unified scheduling framework that encompasses both agents and their tools.

**Efficiency of Context Management.** More efficient mechanisms can be devised to assist context management. For example, the pursuit of time-efficient context management techniques could significantly augment user experience by expediting the processes of context snapshotting and restoration. Also, context compression techniques can also be leveraged prior to snapshotting, which can yield a more space-efficient solution.

**Optimization of Memory and Storage Architecture.** In the context of agent collaboration and communication, the future design of memory and storage systems can adopt a shared approach, enabling the sharing of memory and storage between agents. Such an architecture would enable agents to access a communal pool of memory and storage, thereby improving the agents' decision-making ability since one agent can benefit from other agents' memory or storage. Moreover, future work can explore hierarchical storage solutions, designed to optimize data retrieval and storage efficiency. This could involve prioritizing quicker access and reduced storage allocation for frequently accessed data, and vice versa for less frequently accessed information.

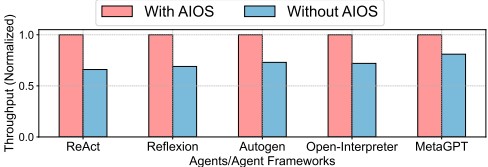 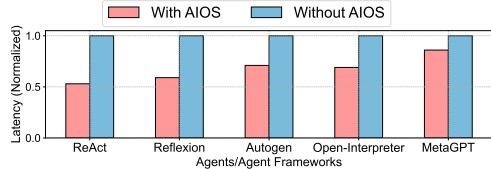

(a) Normalized throughput. Higher is better.      (b) Normalized latency. Lower is better.

Figure 15: Efficiency analysis on different agent frameworks evaluated on the Mistral-7b model on the SWE-Bench-Lite benchmark.

**Safety and Privacy Enhancements.** The aspect of safety in AIOS necessitates protective measures against various attacks, ensuring the system's resilience against malicious attacks, such as jailbreaking of LLM or unauthorized access of other agents' memory. In the realm of privacy, the exploration of advanced encryption techniques is vital for safeguarding data transmission within AIOS, thus maintaining the confidentiality of agent communications. Furthermore, the implementation of watermarking techniques could serve to protect the intellectual property of agent developers by embedding unique identifiers in outputs, facilitating the tracing of data lineage.

In a nutshell, AIOS stands as a motivating body of work that brings a broad spectrum of research opportunities. Each outlined direction not only can build upon the foundational elements of AIOS but also can contribute to the advancement of the field at large.

