# OpenReview forum: "AIOS: LLM Agent Operating System"
_colmweb.org/COLM/2025/Conference — COLM 2025_

### Official Review · Reviewer_ZN9n · 2025-05-12

**Rating:** 6
**Confidence:** 3
**Ethics Flag:** 1

**Summary:**

The blueprint for a native operating system for artificial intelligence is visionary but incomplete, and will require more stringent technical requirements and broader validation to realise its potential.

**Questions To Authors:**

How AIOS' concepts of “resource isolation” and “scheduling” are irreplaceable with the core mechanisms of traditional distributed systems (e.g. Kubernetes) or LLM service frameworks (e.g. Ray)?

**Reasons To Accept:**

The paper presents an interesting direction for resource management for LLM-based agents.

Experimental results suggest that AIOS can improve execution speed for serving agents.

**Reasons To Reject:**

1. The paper mentions FIFO and RR scheduling algorithms. These are standard operating system scheduling algorithms. It is unclear if and how these algorithms accommodate the specific nuances of LLM agent requests (e.g., different computational costs of LLM calls, dependencies between agent tasks, and heterogeneous computational requirements such as dynamic batching, priority preemption).

2. There is no in-depth discussion of practical feasibility, overheads, and compatibility issues (e.g., storage overheads, recovery accuracy) between different LLM architectures (especially closed-source architectures that do not have direct access to logit).

3. The memory manager handles agent interaction history and uses the LRU-K eviction policy. While managing dialogue history is important, it is not entirely clear how this differs significantly from the complex caching and memory management techniques already employed at the application or middleware layers.

4. No mention of robustness issues like security mechanisms, whether debugable, fault tolerance, and the overhead, except for ‘access control’, which is risky to put into a production environment.

---

> ### Author Response · Authors · 2025-06-03
> **Response to Review ZN9n (Part 1/2)**
>
> > ***LLM-nuanced scheduling design***
>
> Our original design philosophy prioritized architectural simplicity to demonstrate the core AIOS framework using standard FIFO and Round Robin algorithms. However, AIOS's modular architecture can easily enable it to incorporate more sophisticated LLM-nuanced scheduling modules. To address the compability concerns of LLM-nuanced scheduling in AIOS, we design a latency-aware scheduler that leverages historical query statistics, specifically token usage patterns. When a new request arrives, the scheduler retrieves top-k similar requests based on cosine similarity and averages their token usage to estimate the computational latency for the current request. The scheduler then prioritizes requests with lower expected latency and assign these requests to the more idle LLM core by tracking the number of in-process requests on each LLM core.
>
> We deploy two identical LLM instances (Llama-3.1-8b) in our evaluation, where the latency-aware analyzes incoming requests based on estimated token usage, with a maximum of 250 agents running concurrently in the system. This setup allows the scheduler to distribute load more effectively. Specifically, we use the HumanEval benchmark and sample 50% of the questions as historical data to store in a vector database and use the rest for evaluation and set top-k as top-5.
>
> | Configuration | Setup | Throughput (req/s) ↑ | Latency (s) ↓ |
> |---------------|-------|---------------------|---------------|
> | **2 GPUs** | w/o AIOS | 0.30 | 4.5 |
> | **2 GPUs** | w/ AIOS | 0.60 | 1.3 |
> | **2 GPUs** | w/ AIOS + Latency-aware scheduling | 0.69 | 1.1 |
>
> The latency-aware scheduling achieves 15% additional throughput improvement and 15% latency reduction compared to basic AIOS scheduling. This demonstrates that AIOS can effectively integrate LLM-specific optimizations.
>
> > ***Discussion of Robustness/Compability Issues***
>
> To demonstrate AIOS's compatibility across different LLM architectures, we conducted additional experiments with three diverse models: GPT-4o-mini (closed-source API), Gemini-1.5-flash (closed-source API), and Llama-3.1-8b (open-source). Since closed-source LLMs cannot save logits for context switching, we adopt streaming APIs and save intermediate generated texts instead to enable snapshot and restore functionalities.
>
> | LLM Configuration | Setup | Throughput (req/s) | Latency (s) |
> |-------------------|-------|-------------------|-------------|
> | **Mixed 3 LLMs** | w/o AIOS | 0.35 | 3.8 |
> | **Mixed 3 LLMs** | w/ AIOS | 0.55 | 1.6 |
> | **Mixed 3 LLMs** | w/ AIOS + Latency-aware scheduling | 0.66 | 1.3 |
>
> The results show that AIOS maintains consistent performance improvement with hybrid LLM cores, demonstrating that our design can easily be extended to combine both open-source and closed-source models through standardized API interfaces.
>
> Regarding robustness, AIOS provides several defensive mechanisms including input validation for all agent requests and comprehensive error handling at each processing stage. More discussions about the security mechanisms will be included in the discussions in our revised version. While we acknowledge that sophisticated recovery mechanisms for enhanced fault tolerance are important, we still would like to focus on the overall AIOS architecture in this paper and would consider more complex security features and recovery mechanisms as important future work.

---

> > ### Author Response · Authors · 2025-06-03
> > **Response to Review ZN9n (Part 2/2)**
> >
> > > ***Design necessity of memory management at the system-level instead of application-level***
> >
> > We acknowledge that our memory management algorithms do not fundamentally differ from existing application/middleware solutions. However, the architectural significance lies in moving agent memory management from application-level to system-level. This architectural change enables AIOS's memory manager to have the potential of making eviction decisions based on global system load rather than isolated individual agent demands. As we would like to maintain a relatively simple memory design in this work to provide the necessary foundation in this first AIOS paper, we consider exploring more sophisticated collaborative memory management mechanisms in our following series of papers.
> >
> >
> > > ***Significance of the AIOS abstraction its relationship with traditional systems/services***
> >
> > We appreciate this insightful question about the irreplacability of AIOS to existing distributed systems (e.g., Kubernetes) or LLM services (e.g., Ray). AIOS operates at a higher level of abstraction specifically designed for agent workloads, and is fundamentally complementary rather than competing with traditional distributed systems like Kubernetes and LLM service frameworks like Ray.
> >
> > While Kubernetes manages containers and Ray manages jobs, AIOS manages agent requests from the agent's perspective, treating LLM, memory, and tool usage as first-class resources. Since the complexity of agents is increasing rapidly [1,2], we think this agent-specific abstraction as critically important to reduce the management complexity of agents as well, which also has been recognized by some recent works [3,4]. We believe this abstraction represents a necessary evolution toward agent-specific environments that can leverage semantic information about agent behavior and requirements to provide more intelligent resource management than general-purpose orchestration systems.
> >
> > [1] Wu et al. AutoGen: Enabling Next-Gen LLM Applications via Multi-Agent Conversation. COLM 2024
> >
> > [2] Hong et al. MetaGPT: Meta Programming for A Multi-Agent Collaborative Framework. ICLR 2024
> >
> > [3] Park et al. Generative Agents: Interactive Simulacra of Human Behavior. 2023
> >
> > [4] Parker et al. MemGPT: Towards LLMs as Operating Systems. 2023

---

> > > ### Comment · Reviewer_ZN9n · 2025-06-09
> > >
> > > Thanks for providing these additional details. It solves my main concerns. I will raise my score accordingly.

---

### Official Review · Reviewer_4vNH · 2025-05-12

**Rating:** 7
**Confidence:** 2
**Ethics Flag:** 1

**Summary:**

This paper introduces the AIOS framework, designed to enhance the parallel efficiency of multiple LLM-based agents. By isolating the resources of LLM services within a dedicated AIOS kernel, the framework achieves up to a 2.1× improvement in efficiency. Additionally, it provides an SDK to improve usability and ease of integration.

The paper clearly outlines the architecture of AIOS and presents experimental results across various agent frameworks. The proposed system significantly improves concurrency and overall task efficiency for agent-based applications.

**Questions To Authors:**

1. It looks like each component in the AIOS architecture has a special design. I wonder if there is an ablation study that shows that the efficiency improvement is credited more to certain designs
2. What is the experimental setup for the legend without AIOS on the bar charts? How is the LLM service call scheduled in this baseline?
3. How does Context Manager ensure efficient snapshot and restore
operations? These operations seem to be taking up a lot of cache and time in the case of multiple concurrency.

**Reasons To Accept:**

1. The framework demonstrates superior efficiency in high-concurrency settings, achieving higher throughput and lower latency.

2. Validated across various agent tasks, the framework maintains comparable task performance to traditional systems while offering faster execution speed.

3. An SDK is provided to streamline integration and improve ease of use.

**Reasons To Reject:**

1. Lack of experimental comparison with classical frameworks (e.g., Autogen, Langchain).

---

> ### Author Response · Authors · 2025-06-02
> **Response to Reviewer 4vNH**
>
> > ***Comparison with running with traditional agent frameworks***
>
> We would like to clarify that our evaluation indeed **includes** comparison with classical frameworks. Specifically, we implemented agents using frameworks, i.e., **Autogen, Open-Interpreter and MetaGPT**. The "w/o AIOS" baseline directly uses the **concurrent execution mechanisms provided by these frameworks themselves**. This ensures our comparison is straightforward to reflect performance improvements of using AIOS over the standard concurrent execution patterns used in established agent frameworks.
>
> > ***More ablation studies***
>
> We appreciate this important question about the individual contributions of AIOS modules. Each module in AIOS serves a critical role analogous to traditional OS components, specifically adapted for agent environments. We conducted ablation studies examining the impact of removing key modules:
>
> | Configuration | Throughput (req/s) ↑ | Latency (s) ↓ |
> |---------------|---------------------|---------------|
> | Full AIOS | 0.28 | 3.0 |
> | w/o Memory Manager | 0.23 (-18%) | 3.5 (+17%) |
> | w/o Tool Manager | 0.22 (-21%) | 3.6 (+20%) |
> | w/o Storage Manager | 0.25 (-11%) | 3.3 (+10%) |
>
> The ablation results show that each module contributes meaningfully to system performance. The Memory Manager improves throughput by reducing redundant conversation history loading. The Tool Manager provides the largest benefit by preventing error-recovery cycles when multiple agents access tools concurrently. The Storage Manager primarily affects persistent data operations.
>
> > ***Details of experimental setting regarding w and w/o AIOS***
>
> To ensure fair comparison, we use identical prompts and the same LLM backend infrastructure when executing agents with and without AIOS, with the key difference being the scheduling strategy. In the baseline w/o AIOS, LLM service calls are managed by the native concurrent mechanism provided by the agent frameworks, competing for GPU resources without coordination. In contrast, AIOS employs the scheduling that pre-allocates GPU resources to a limited number of concurrent requests, preventing resource contention and OOM conditions.
>
> > ***Design rationale of context switch***
>
> We acknowledge that context switching introduces computational overhead, which is an inherent trade-off in any multi-tasking system. AIOS mitigates this overhead by caching logits at the last layer, allowing generation to resume from cached logits rather than restarting the entire computation. This design choice ensures our optimization is orthogonal to KV cache reuse approaches like vLLM[1] and SGLang[2], enabling AIOS to leverage their advantages and work jointly with existing optimizations. While context switching does introduce latency overhead similar to traditional OS context switches, this cost is justified by preventing system-wide blocking scenarios where long-running requests monopolize resources. Our experimental results demonstrate that despite the context switching overhead, AIOS still achieves superior throughput and latency compared to systems without scheduling, confirming that switching costs are acceptable.
>
> [1] Kwon et al. Efficient memory management for large language model serving with paged attention. SOSP 2023.
>
> [2] Zheng, Lianmin, et al. Sglang: Efficient execution of structured language model programs. NeurIPS 2024.

---

> > ### Comment · Reviewer_4vNH · 2025-06-04
> > **Thank you for reply!**
> >
> > The authors have addressed all my concerns thoroughly and have done an excellent job. I will not raise my score further as it is already high—good luck!

---

### Official Review · Reviewer_FGa7 · 2025-05-15

**Rating:** 8
**Confidence:** 3
**Ethics Flag:** 1

**Summary:**

This paper proposes the architecture of AIOS in the context of managing LLM-based agents.

The primary contribution of this paper is the design and implementation of AIOS, a novel operating system-like architecture specifically tailored for managing LLM-based agents.

**Questions To Authors:**

You mention treating each LLM instance as a “core.” How does the system handle load balancing and dynamic allocation among these cores?

The benchmarks used represent structured tasks. Do they reflect real-world scenarios of intensive, continuous multi-agent use? What limitations do these benchmarks impose on evaluating scalability?

**Reasons To Accept:**

The work is original. While concepts like treating LLMs as operating systems have been explored, AIOS is the first (to my knowledge) to formalize and implement a comprehensive OS-like system with a dedicated kernel and SDK for concurrent LLM-based agent management.

The experimental results are competitive. Benchmarks across multiple frameworks and tasks show that AIOS maintains or improves performance while enhancing throughput (up to 2.1×) and reducing latency.

**Reasons To Reject:**

The implementation details in the appendix are extensive and might be better summarized in the main text.

Real-world deployment scenarios (e.g., user studies or production cases) are not discussed.

---

> ### Author Response · Authors · 2025-06-02
> **Response to Reviewer FGa7**
>
> > ***Summarize implementation details in main text***
>
> Thank you for your recognition of our extensive implementation details. We will summarize them accordingly in the main text in the revised version.
>
> > ***Overhead balancing between different LLM cores***
>
> We conduct further evaluations of our architecture with multiple LLM cores on multiple GPUs to demonstrate AIOS's load balancing capabilities. The experimental setup is as follows: Each GPU hosts one LLM instance (core) running Llama-3.1-8b, with a maximum of 250 agents running concurrently in the system. The rest of the experimental settings remain consistent with our ablation study in Appendix D.1.
>
> To handle overhead balancing across multiple LLM cores, we implement a simple yet effective round-robin strategy that monitors the number of in-process requests for each LLM core and dynamically assigns incoming requests to the least-loaded core. This approach ensures balanced resource utilization and prevents any single core from becoming a bottleneck.
>
> | GPU Configuration | Setup | Throughput (req/s) ↑ | Latency (s) ↓ |
> |-------------------|-------|---------------------|---------------|
> | **2 GPUs** | w/o AIOS | 0.30 | 4.5 |
> | **2 GPUs** | w/ AIOS | 0.60 | 1.3 |
> | **4 GPUs** | w/o AIOS | 0.55 | 2.4 |
> | **4 GPUs** | w/ AIOS | 1.16 | 0.7 |
>
> The results demonstrate AIOS's effective overhead balancing across multiple LLM cores. As the number of GPUs increases, AIOS consistently achieves better throughput (2× improvement on 2 GPUs, 2.1× on 4 GPUs) and lower latency (71% reduction on both configurations) compared to baseline approaches. This consistent performance advantage across different scales shows that AIOS's load balancing mechanism successfully distributes computational overhead across multiple LLM cores, thus benefiting efficient execution of agents in complex multi-LLM deployment scenarios.
>
> > ***Discussion of limitations on evaluations of existing benchmarks in continous agent-use scenarios***
>
> While we have adapted existing benchmarks to simulate concurrent agent usage, we acknowledge that these benchmarks cannot fully capture long-term or intensive, continuous multi-agent use patterns. We appreciate you highlighting this limitation of current evaluations. However, we think benchmarking long-term agent-use is not trivial and we plan to include this discussion in our revised version to encourage community efforts to developing benchmarks towards continuous agent-use scenarios.
>
> > ***Discussion of real-world deployment and user studies***
>
> We agree that real-world user studies are crucial for validating AIOS's effectiveness. We have been developing a deployed version of AIOS and are conducting early-stage real-world testing. However, this deployment remains in its initial phase and requires more extensive evaluation before drawing meaningful conclusions about production performance. We would like to include this in the discussion of our revised version and consider comprehensive production-use evaluation and user studies as our important future work to provide more valuable insights.

---

> > ### Comment · Reviewer_FGa7 · 2025-06-09
> >
> > The authors have addressed all my concerns

---

### Official Review · Reviewer_UuGH · 2025-05-25

**Rating:** 7
**Confidence:** 3
**Ethics Flag:** 1

**Summary:**

- The paper targets the lack of formal scheduling and resource-isolation support in today’s LLM-agent frameworks, where agents can overload the GPU or external tools.
- It proposes AIOS, a three-layer architecture. (1) A core layer that handles scheduling (deciding which agent runs when). (2) Tools that manage memory, storage, and access to resources. (3) An interface that lets AI agents use these tools easily.
- Experiments on five agent frameworks show equal or better benchmark success rates and up to 2.1× higher throughput plus lower average latency on a single RTX-A5000.

**Reasons To Accept:**

- The work addresses a clear, real-world bottleneck: many concurrent agents fighting for one LLM instance or tool chain.
- System design is well documented. The appendix includes syscall catalogs, adapter APIs, and pseudocode.
- Evaluation covers closed-source (API) and open-source (local) models,  and 4 benchmarks (HumanEval, MINT, GAIA, SWE-Bench-Lite).

**Reasons To Reject:**

- The baseline “w/o AIOS” is vaguely defined. Please clarify whether it uses (i) the same prompts, (ii) the same scheduling method, and (iii) the same backend for LM inference.
- The paper only includes ablation of the scheduler. I wonder how other part in the system influence the latency and throughput.
- In Figure 8, I understand that with poor scheduling, the agent waiting time can increase roughly linearly as the number of agents increases. However, once GPU utilization reaches 100%, I would expect the total execution time to become less sensitive to the scheduling strategy. Could you clarify why the results in the figure still show differences in execution time under 2K agent?
- All tests use a single GPU. It would be helpful to discuss how the AIOS scales with multiple GPUs or distributed LLM.

---

> ### Author Response · Authors · 2025-06-02
> **Response to Reviewer UuGH**
>
> > ***Details of experimental setting regarding w and w/o AIOS***
>
> We appreciate this important clarification request. The comparison setting is as below: **Same prompts:** Both AIOS and the baseline w/o AIOS use identical prompts to ensure fair comparison. **Same backend for LM inference:** Both configurations use the identical LLM backend infrastructure to isolate the impact of scheduling strategy. **Different scheduling methods:** The "w/o AIOS" baseline relies on OS-level scheduling, where each LLM query can preemptively access GPU resources without coordination. In contrast, AIOS's scheduling mechanism is to pre-allocate GPU resources to a limited number of concurrent LLM queries to prevent recovery overheads from OOM.
>
> > ***Functionalities of other modules in AIOS***
>
> We appreciate this important question about the individual contributions of AIOS modules. While our paper focused primarily on scheduler ablation, each module in AIOS serves a critical role analogous to traditional OS components, specifically adapted for agent environments. We conducted additional ablation studies examining the impact of removing key modules. The setting is the same as introduced in ablation study of scheduler in Appendix D.1 of our paper.
>
> | Configuration | Throughput (req/s) ↑ | Latency (s) ↓ |
> |---------------|---------------------|---------------|
> | Full AIOS | 0.28 | 3.0 |
> | w/o Memory Manager | 0.23 (-18%) | 3.5 (+17%) |
> | w/o Tool Manager | 0.22 (-21%) | 3.6 (+20%) |
> | w/o Storage Manager | 0.25 (-11%) | 3.3 (+10%) |
>
> Through the ablation studies, we find that memory manager helps improve throughputs due to its reduction of the length conversation histories. Storage manager works similarly as memory manager but it can only affect persistent data handling scenarios. The tool manager contributes as it prevents the error-then-recover process when multiple agents attempt concurrent tool usage. These modules work jointly to improve the execution efficiency of agents on AIOS.
>
> > ***Scheduling effectiveness when scaling up***
>
> Thank you for your insightful question, we would like to clarify that even at 100% GPU utilization, scheduling strategy remains critical for the following reasons: **GPU Memory Management Overhead:** Poor scheduling causes frequent memory allocation/deallocation as agents compete for GPU memory. **OOM Prevention:** At high agent counts, uncoordinated GPU access frequently triggers OOM conditions, forcing expensive garbage collection and memory reorganization that increases execution time despite high utilization. However, AIOS adopts the pre-allocation strategy to assign GPU access to a limited number of concurrent requests.
> The key insight is that **preventing failures is exponentially more efficient than recovering from them.** As agent count increases, AIOS prevents the exponential degradation that occurs when resource contention triggers system-level failures and this explains why the performance gap grows larger rather than smaller at high utilization.
>
> > ***Evaluations using multiple GPUs & LLMs***
>
> We conduct further evaluations of our architecture with multiple LLM cores on multiple GPUs and the setup is as below: Each GPU hosts one LLM instance(core) (Llama-3.1-8b), and there can be a maximum of 250 agents running concurrently in the system. The rest of the experimental settings are kept the same as introduced in our ablation study in Appendix D.1. To handle load balancing across multiple LLM cores, we adopt a simple round-robin strategy that counts the number of in-process requests for each LLM core and assigns incoming requests to the least-loaded LLM core.
>
> | GPU Configuration | Setup | Throughput (req/s) ↑ | Latency (s) ↓ |
> |-------------------|-------|---------------------|---------------|
> | **2 GPUs** | w/o AIOS | 0.30 | 4.5 |
> | **2 GPUs** | w/ AIOS | 0.60 | 1.3 |
> | **4 GPUs** | w/o AIOS | 0.55 | 2.4 |
> | **4 GPUs** | w/ AIOS | 1.16 | 0.7 |
>
> The results show that as number of GPU increases, AIOS can also achieve better throughputs and lower latency, which suggests the effectiveness of AIOS in more complex multi-LLM and multi-GPU scenarios.

---

> > ### Comment · Reviewer_UuGH · 2025-06-09
> >
> > You have addressed all my concerns. I do not have any further questions at this time and will keep my current assessment.

---

### Author Response · Authors · 2025-06-02
**General response**

We thank the reviewers for their constructive comments and the recognition of the novelty and contribution of our work. We hope the following clarfications and addtional experimental results can help address the concerns.

---

### Decision · Program_Chairs · 2025-07-08

**Decision:**

Accept

**Comment:**

This paper introduces the AIOS framework, designed to enhance the parallel efficiency of multiple LLM-based agents. By isolating the resources of LLM services within a dedicated AIOS kernel, the framework achieves up to a 2.1× improvement in efficiency. Additionally, it provides an SDK to improve usability and ease of integration. The paper clearly outlines the architecture of AIOS and presents experimental results across various agent frameworks. The proposed system significantly improves concurrency and overall task efficiency for agent-based applications.

Reasons To Accept:
* The work addresses a clear, real-world bottleneck: many concurrent agents fighting for one LLM instance or tool chain.
* System design is well documented. The appendix includes syscall catalogs, adapter APIs, and pseudocode.
* Evaluation covers closed-source (API) and open-source (local) models, and 4 benchmarks (HumanEval, MINT, GAIA, SWE-Bench-Lite).

Reasons To Reject:
* The baseline “w/o AIOS” is vaguely defined. Please clarify whether it uses (i) the same prompts, (ii) the same scheduling method, and (iii) the same backend for LM inference.
* The paper mentions FIFO and RR scheduling algorithms. These are standard operating system scheduling algorithms. It is unclear if and how these algorithms accommodate the specific nuances of LLM agent requests (e.g., different computational costs of LLM calls, dependencies between agent tasks, and heterogeneous computational requirements such as dynamic batching, priority preemption).
* The paper only includes ablation of the scheduler. I wonder how other part in the system influence the latency and throughput.